# Roboflow100-VL: A Multi-Domain Object Detection Benchmark for Vision-Language Models

Peter Robicheaux[1],*, Matvei Popov[1],*, Anish Madan[2], Isaac Robinson[1],
Joseph Nelson[1], Deva Ramanan[2], Neehar Peri[2]
[1]Roboflow, [2]Carnegie Mellon University

rf100-vl.org

## Abstract

Vision-language models (VLMs) trained on internet-scale data achieve remarkable zero-shot detection performance on common objects like `car`, `truck`, and `pedestrian`. However, state-of-the-art models still struggle to generalize to out-of-distribution classes, tasks and imaging modalities not typically found in their pre-training. Rather than simply re-training VLMs on more visual data, we argue that one should align VLMs to new concepts with annotation instructions containing a few visual examples *and* rich textual descriptions. To this end, we introduce Roboflow100-VL, a large-scale collection of 100 multi-modal object detection datasets with diverse concepts not commonly found in VLM pre-training. We evaluate state-of-the-art models on our benchmark in zero-shot, few-shot, semi-supervised, and fully-supervised settings, allowing for comparison across data regimes. Notably, we find that VLMs like GroundingDINO and Qwen2.5-VL achieve less than 2% zero-shot accuracy on challenging medical imaging datasets within Roboflow100-VL, demonstrating the need for few-shot concept alignment. Lastly, we discuss our recent CVPR 2025 Foundational FSOD competition and share insights from the community. Notably, the winning team significantly outperforms our baseline by 17 mAP! Our code and dataset are available on GitHub and Roboflow.

## 1  Introduction

Vision-language models (VLMs) trained on web-scale datasets achieve remarkable zero-shot performance on many popular academic benchmarks [156, 87, 132]. However, the performance of such foundation models varies greatly when evaluated in-the-wild, particularly on out-of-distribution classes, tasks (e.g. material property estimation, defect detection, and contextual action recognition) and imaging modalities (e.g. X-rays, thermal spectrum data, and aerial imagery). In this paper, we introduce Roboflow100-VL (RF100-VL), a large-scale multi-domain dataset to benchmark state-of-the-art VLMs on hundreds of diverse concepts not typically found in internet pre-training.

**Status Quo.** Foundation models are often trained on large-scale datasets curated from diverse sources around the web. However, despite their scale and diversity, these pre-training datasets still follow a long-tail distribution [121], causing foundation models to generalize poorly to rare concepts [106]. A common approach for improving the performance of VLMs is to scale up training data and model size [24]. However, we argue that some data will always remain out-of-distribution, whether due to being sequestered from the internet or being created after the model's training cutoff [144], motivating the need to learn new concepts from a few examples.

**Evaluating Out-of-Distribution Generalization.** Existing benchmarks primarily assess spatial understanding through visual question answering (VQA) and common sense reasoning [87, 157,

---

*Equal Contribution

39th Conference on Neural Information Processing Systems (NeurIPS 2025) Track on Datasets and Benchmarks.

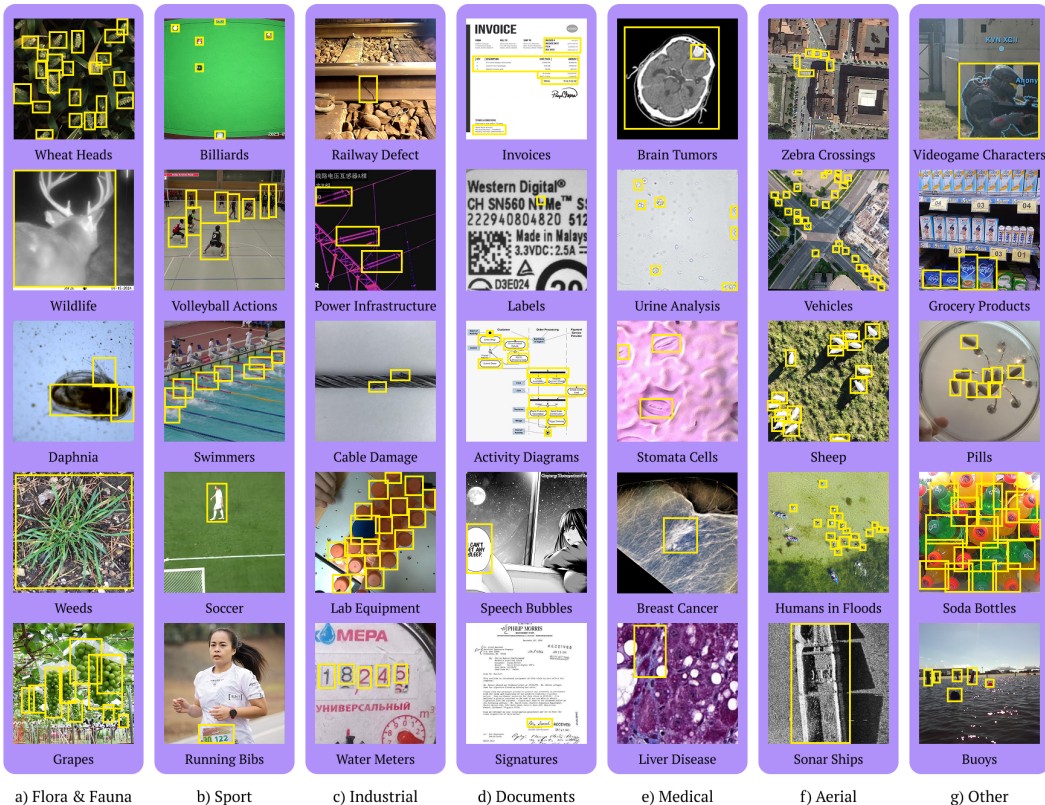

a) Flora & Fauna    b) Sport    c) Industrial    d) Documents    e) Medical    f) Aerial    g) Other

Figure 1: **Roboflow100-VL Dataset.** We identify a set of 100 challenging datasets from Roboflow Universe that contain concepts not typically found in internet-scale pre-training. To simplify analysis, we cluster these 100 datasets using per-dataset CLIP [113] embeddings into seven categories. We visualize examples from each of these categories above. Furthermore, we also generate multi-modal instructions for each dataset with a few visual examples and rich textual descriptions per class to facilitate few-shot concept alignment.

132]. However, we argue that evaluating model performance on compositional reasoning benchmarks alone does not effectively measure generalization to out-of-distribution tasks. Moreover, current spatial understanding and grounding benchmarks (e.g. RefCOCO [155] and OdinW [82]) typically evaluate performance on classes commonly found in internet pre-training. We demonstrate that such benchmarks artificially inflate model performance and are not representative of many real-world applications (cf. Table 1). To address this limitation, we introduce RF100-VL, a large-scale detection benchmark comprised of 100 multi-modal datasets from diverse domains (cf. Fig. 1). Importantly, we carefully curate RF100-VL such that it cannot be solved by simply prompting state-of-the-art models with class names. Specifically, we include datasets where classes are labeled using scientific names (e.g. liver `fibrosis` and `steatosis`), acronyms (e.g. `DIP` and `MCP`), context-dependent names (e.g. detecting a `block` vs. `set` in the context of volleyball), material properties (e.g. `paper` vs. `soft plastic`), and diverse imaging modalities (cf. Fig. 2). We posit that models must leverage multi-modal contextual information (presented in the form of multi-modal annotator instructions) to effectively align to target concepts in RF100-VL.

**Multi-Modal Annotator Instructions.** Annotating large-scale datasets is an iterative process that often requires extensive discussions between data curators and annotators to clarify class definitions and ensure label consistency. These (often multi-modal) labeling instructions provide rich contextual information not provided by class names alone. We argue that aligning foundation models to target concepts can be principally addressed through the lens of few-shot learning by presenting vision-language models with visual examples and rich textual descriptions per class (cf. Fig. 3). Importantly, this approach mirrors how we align human annotators to concepts of interest with few-shot multi-modal examples [34, 91].

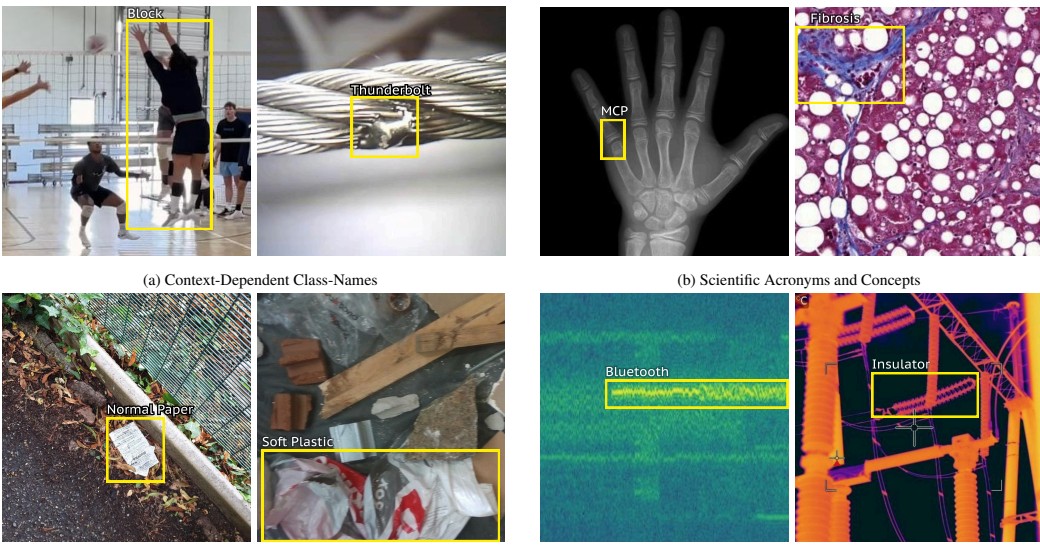

Figure 2: **Hard Examples in Roboflow100-VL.** Our dataset is particularly challenging because it is difficult to detect objects in RF100-VL using class-names alone. Specifically, we select datasets where classes are labeled using scientific names, acronyms, context-dependent names, material properties. We posit that models must leverage multi-modal contextual annotations to address such hard examples.

**Contributions.** We present three major contributions. First, we introduce RF100-VL, a large-scale, multi-domain benchmark designed to evaluate vision-language models (VLMs) on challenging real-world use cases. We evaluate state-of-the-art models on our benchmark in zero-shot, few-shot, semi-supervised, and fully-supervised settings, allowing for comparison across data regimes. Our extensive experiments highlight the difficulty of adapting VLMs to out-of-distribution tasks and reveal the limitations of current state-of-the-art methods. Lastly, we highlight the results of our recent CVPR 2025 challenge hosted in conjunction with the Workshop on Visual Perception via Learning in An Open World.

## 2 Related Works

**Vision Language Models** are trained using large-scale, weakly supervised image-text pairs sourced from the web. Although many VLMs primarily focus on classification [113] or image understanding, recent methods address spatial understanding with open-vocabulary detectors. Early approaches adapted VLMs for object detection by classifying specific image regions [62, 63] or integrating detection components into frozen [78] or fine-tuned [98, 97, 52] encoders. In contrast, RegionCLIP [160] employs a multi-stage training strategy that involves generating pseudo-labels from captioning data, performing region-text contrastive pre-training, and fine-tuning on detection tasks. GLIP [83] treats detection as a phrase grounding problem by using a single text query for the entire image. Detic [161] improves long-tail detection performance by utilizing image-level supervision from ImageNet [117]. Notably, recent VLMs achieve remarkable zero-shot performance and are widely used as "black box" models in diverse downstream applications [90, 108, 75, 103, 130]. More recently, multi-modal large language models (MLLMs) such as Qwen2.5-VL [28] and Gemini 2.5 Pro [47] frame spatial understanding as a text generation task. Interestingly, such generalist MLLMs perform worse at object detection than task-specific models like GroundingDINO [86]

**Fine-Tuning Vision-Language Models** is crucial for adapting foundation models to downstream tasks [68, 158, 59]. Traditional fine-tuning methods, such as linear probing [37, 67] and full fine-tuning [146, 151] can be computationally expensive. Instead, parameter-efficient approaches like CLIP-Adapter [59] and Tip-Adapter [159] optimize lightweight MLPs while keeping encoders frozen. Although prior few-shot learners commonly used meta-learning [154], more recent approaches show that transfer learning generalizes better to novel categories [145]. In particular, Pan et. al. [105] demonstrates that transfer learning can be effectively used to fine-tune foundation models using

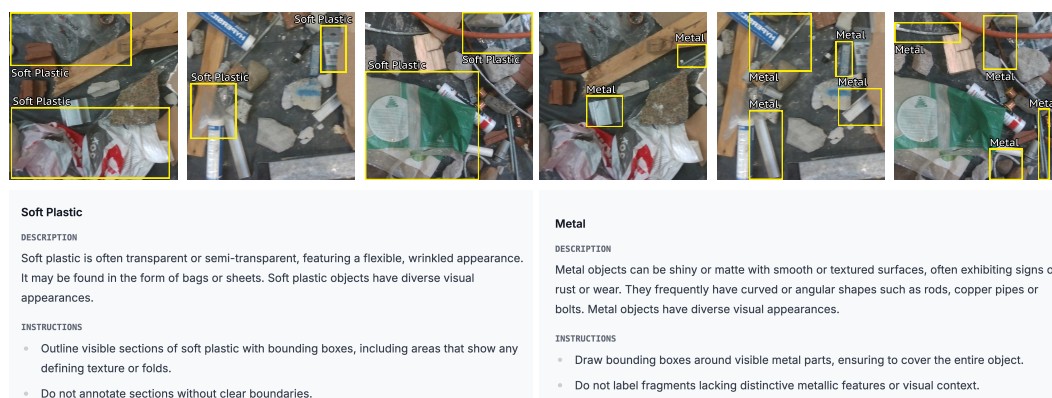

Figure 3: **Multi-Modal Few-Shot Examples.** We present an example of the few-shot visual examples and rich text descriptions used for in-context prompting and fine-tuning. Notably, image examples used for each class may overlap and are only guaranteed to have exhaustive annotations for one class. Such multi-modal examples help clarify ambiguous concepts like `soft plastic` and `metal`.

a few multi-modal examples. More recently, in-context learning [152] demonstrates promising results for test-time few-shot adaptation without gradient-based fine-tuning. We explore such test-time fine-tuning strategies in the context of MLLMs [47, 28] to learn from multi-modal annotator instructions.

**Benchmarking Vision-Language Models** is of significant interest to the community. State-of-the-art VLMs are typically evaluated using benchmarks such as MMStar [35], MMMU [157], MME [84], ScienceQA [89], MMBench [87], MM-Vet [156], Seed-Bench [81], and MMVP [133]. These benchmarks evaluate a broad set of vision-language tasks, including fine-grained perception, reasoning, common sense knowledge, and problem solving in various domains. However, existing evaluations primarily focus on multi-modal understanding in the context of visual question answering (VQA). In contrast, RF100-VL evaluates VLM detection accuracy given a few visual examples and rich textual descriptions. Prior VLM grounding benchmarks like RefCOCO [155] often focus on referential grounding of common object categories. Recent efforts like ODinW [82] consider more challenging scenarios by sourcing real-world data from Roboflow [38]. However, we find that state-of-the-art methods achieve high zero-shot accuracy on RefCOCO and OdinW [28], suggesting that these datasets may not be well suited for evaluating foundational few-shot object detection [91].

## 3 Roboflow100-VL Benchmark

As shown in Fig. 1, RF100-VL consists of diverse datasets not typically found in internet-scale pre-training. We highlight our data curation procedure (Section 3.1) and present several baselines to evaluate state-of-the-art models in the zero-shot and few-shot settings (Section 3.2). We also evaluate models under the semi-supervised and fully-supervised settings in Appendix F.

### 3.1 Creating Roboflow100-VL

We source our datasets from Roboflow Universe, a community-driven platform that hosts diverse open-source datasets created to solve real-world computer vision tasks. With more than $500,000$ public datasets spanning medical imaging, agriculture, robotics, and manufacturing, we focus on selecting high-quality datasets not commonly found in internet-scale pre-training (e.g. COCO [85], Objects365 [124], GoldG [73], CC4M [125]) to better assess VLM generalization to rare concepts. When selecting candidates for RF100-VL, we prioritized datasets where images contained multiple objects, ensuring more realistic evaluation beyond classification. In addition, we sought out datasets with semantically ambiguous names (e.g. "button" can refer to both clothing and electronics) to encourage algorithms to leverage multi-modal annotator instructions rather than simply relying on class names. We manually validate the labeling quality of each dataset to ensure exhaustive

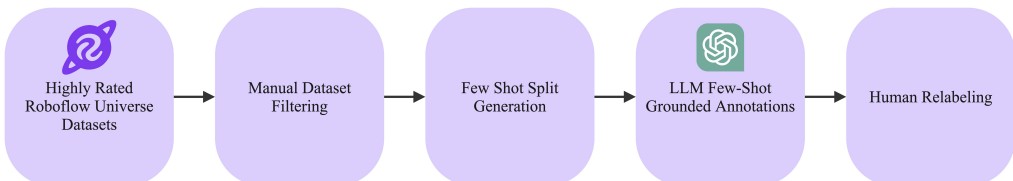

Figure 4: **Dataset Curation.** We begin by sorting all object detection datasets on Roboflow Universe by stars as a proxy for quality and usefulness to the community. Next, we manually filter out all datasets with common classes, datasets where images only have a single focal object, or datasets with watermarks. We generate 10-shot splits following the protocol defined by Wang et.al. [145], where we find a subset of images with 10 total instances per class. We use these 10-shot splits to generate visually grounded "annotator instructions", and manually update these instructions to add any salient details missed by GPT-4o. Finally, human labelers verify that all images within a dataset follow consistent annotation policies (e.g. bounding-box fit, semantic legibility of class names, and completeness of annotation instructions).

annotations. In cases without exhaustive annotations, we manually re-annotate the dataset to the best of our ability (cf. Fig 4). In total, we spent 1693 hours labeling, reviewing, and preparing the dataset.

**Multi-Modal Annotation Generation.** Annotator instructions offer precise class definitions and visual examples that help clarify annotation policies (e.g. by highlighting typical cases, corner cases, and negative examples) and improve labeling accuracy. Despite providing significant value during the labeling process, few datasets publicly release these annotator instructions. Recognizing the importance of these instructions in aligning humans with target concepts of interest, we generate multi-modal annotator instructions for all 100 datasets within RF100-VL (cf. Fig 3).

We prompt GPT-4o [24] to generate an initial set of annotator instructions, providing in-context examples based on the nuImages annotator guidelines. Our prompt includes a structured output template, along with dataset metadata, class names, and few-shot visual examples per class. In practice, we find that GPT-4o often overlooks the few-shot images and instead relies heavily on class names to generate class descriptions. Notably, GPT-4o struggles when class names are uninformative and sometimes produces overly vague instructions that, while correct, lack useful detail. To address this, we manually verify all generated annotator instructions to mitigate hallucinations and incorporate additional informative visual details missed by the model. We include our annotation generation prompt in Appendix P.

**Dataset Statistics.** Figure 5 (left) presents an overview of the different types of datasets within RF100-VL, detailing the number of classes, images and annotations per category. RF100-VL contains a total of 564 classes and 164,149 images, with over 1.3 million annotations. The "Other" category has the highest number of classes (142), followed by "Industrial" (122) and "Flora & Fauna" (70). Despite having fewer classes, the "Flora & Fauna" category has the highest number of images (46,718) and annotations (441,677), indicating a higher density of annotations per image. Figure 5 (right) provides a visual representation of class distribution, reinforcing the dominance of the "Other", "Industrial", and "Flora & Fauna" categories. In contrast, "Sports" has the fewest classes (36) and the least representation in RF100-VL. Despite consisting of 100 datasets, RF100-VL has about half the number of images as COCO [85], making this an approachable benchmark for the academic community.

## 3.2 State-of-the-Art Baselines

We train and evaluate all models on each dataset within RF100-VL independently. Importantly, we do not tune any parameters or modify zero-shot prompts per-dataset. For all models, we compute metrics using pycocotools with maxDets set to 500 instead of the usual 100 because there are many images with more than 100 objects. We discuss our evaluation protocol further in Appendix B.

**Zero-Shot Baselines** prompt models with class names or expressive descriptions [96] to detect target concepts. However, the effectiveness of zero-shot prompting depends on the pre-training data: If the target class name is semantically meaningful and aligns well with the model's foundational

| Dataset Type | # Classes | # Images | # Anno. |
|--------------|-----------|----------|---------|
| Aerial | 29 | 11,627 | 186,789 |
| Document | 88 | 21,418 | 127,129 |
| Flora & Fauna | 70 | 46,718 | 441,677 |
| Industrial | 122 | 29,758 | 205,627 |
| Medical | 77 | 16,369 | 125,433 |
| Sports | 36 | 8,443 | 58,508 |
| Other | 142 | 29,816 | 210,328 |
| All | 564 | 164,149 | 1,355,491 |

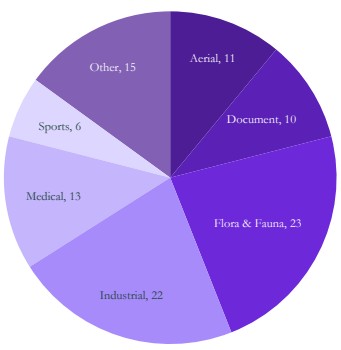

Figure 5: **Dataset Statistics.** The table on the left provides details on the number of classes, images, and annotations across different dataset types within RF100-VL. The figure on the right illustrates the distribution of dataset types by count. Notably, despite containing 100 datasets, RF100-VL is 50% the size of COCO [85] (by number of images) and can feasibly be benchmarked on academic-level compute.

pre-training, performance is strong; otherwise, the model fails catastrophically. We benchmark the zero-shot performance of Detic [161], OWLv2 [97], GroundingDINO [86], MQ-GLIP [152], QwenV2.5-VL [28] and Gemini 2.5 Pro [47].

**Few-Shot Baselines.** We evaluate three types of few-shot baselines: visual prompting, multi-modal prompting, and federated fine-tuning. Visual prompting uses images of target concepts that are difficult to describe through text as prompts to help models learn novel concepts in-context. For example, while "hard plastic" is a broad and ambiguous category that is hard to define through text, providing image examples improves concept alignment. Typically, visual prompts are tokenized and fed as inputs to a frozen VLM. Here, we apply MQ-GLIP [152] with image prompting. Multi-modal prompting combines language and visual prompts to leverage multi-modal features. Intuitively, using both text and images yields better alignment than using either modality alone. In the case of "soft plastic", ambiguous concepts can be clarified with textual descriptions (e.g., "thin plastic film" and "plastic bag") alongside visual examples. Both visual and language prompts are tokenized and separately fed into a frozen VLM. We evaluate MQ-GLIP [152], and Gemini 2.5 Pro [47] by prompting models with class names, few-shot images, and annotator instructions. Lastly, federated fine-tuning modifies the standard cross-entropy classification to only treat exhaustively annotated classes as true negatives for each image. We follow the implementation from Madan et. al. [91] when fine-tuning Detic [161]. We slightly modify the federated loss when fine-tuning YOLO [71, 74] to avoid using Madan et. al's frequency prior, opting to instead determine hard negatives using per-image annotations.

## 4 Experiments

We conduct extensive experiments to evaluate the performance of state-of-the-art models on RF100-VL. We present our zero-shot and few-shot results below. See Appendix A for additional implementation details and Appendix F for semi-supervised and fully supervised results.

### 4.1 Metrics

Each dataset within RF100-VL is independently evaluated using AP. We report the average accuracy per super-category to simplify analysis. RF100-VL includes datasets that are out-of-distribution from typical internet-scale pre-training data, making it particularly challenging (even for VLMs). To construct the few-shot split, we follow the $K$-shot dataset creation process established by [145]. See Appendix E for further discussion on few-shot split selection. Importantly, all methods across data regimes are evaluated on the same fully annotated test set. In Table 1, we highlight that prior methods report different results on COCO and OdinW than our reproduced results. YOLOv8 [71] and YOLOv11 [74] achieve slightly different performance on COCO because the original results are reported using Ultralytics, whereas our results are computed using pycocotools. Importantly,

this discrepancy in tooling yields a larger disparity on RF100-VL, discussed further in Appendix B. Further, we find that Qwen2.5-VL evaluates on ODinW using a referential grounding protocol (reported, see GitHub issue) instead of a traditional object detection protocol (ours). Specifically, referential grounding prompts a model with only the true positive classes in each test image, while object detectors prompt a model with *all* classes. The former dramatically reduces the number of false positives. We evaluate Gemini 2.5 Pro using both protocols for completeness.

## 4.2 Empirical Analysis of Results

**State-of-the-Art Zero-Shot and Few-Shot Models Struggle on Roboflow100-VL.** RF100-VL is a much harder dataset than prior open-vocabulary object detection benchmarks. Specifically, GroundingDINO achieves 49.2 mAP on ODinW-13, but only reaches 15.7 mAP on RF100-VL. Similar trends can be seen with Qwen2.5-VL and Gemini 2.5 Pro (cf. Table 1). Notably, both RF100-VL and ODinW-13 are sourced from Roboflow Universe, but our dataset is carefully curated to evaluate performance on target concepts not typically found in internet-scale pre-training.

**Open-Vocabulary Object Detectors Outperform MLLMs.** We find that open-vocabulary object detectors like Detic, GroundingDINO, OWLv2, and MQ-GLIP consistently outperform MLLMs like Qwen 2.5 VL, Gemini 2.5 Pro, despite these MLLMs pre-training on orders of magnitude more data. We posit that this poor performance can be attributed to MLLMs not reporting per-box confidence scores or ensuring that predictions don't overlap (e.g. non-maximal suppression). This highlights the advantage of task-specific architectures over generalist models.

**Multi-Modal Annotator Instructions Provide Limited Benefit.** Somewhat surprisingly, state-of-the-art MLLMs struggle to benefit from multi-modal annotator instructions. In fact, prompting with instructions provides inconsistent benefit compared to prompting with class names (e.g. Qwen2.5VL improves but Gemini 2.5 Pro degrades considerably). Intuitively, we expect annotator instructions to improve object detection performance by resolving semantic ambiguity in class names and providing rich contextual information. However, we posit that this performance decline can be attributed to the fact that MLLMs are instruction-tuned for open vocabulary detection with rigid prompt structures, making it difficult to effectively leverage additional contextual information.

**Large-Scale Pre-Training Improves Fine-Tuned Few-Shot Performance in Specialists.** We find that fine-tuning GroundingDINO [83] achieves the best few-shot performance, significantly outperforming all YOLO variants by more than 10%. Notably, all gradient-based fine-tuning baselines outperform in-context visual prompting and multi-modal prompting methods, suggesting that in-context prompting provides limited benefit for rare classes not seen in pre-training. We posit that GroundingDINO's large-scale task-specific pre-training makes it easier to learn new concepts during fine-tuning.

Table 1: **Comparison to Other Benchmarks.** We find that state-of-the-art MLLMs achieve considerably lower performance on RF100-VL compared to OdinW-13, highlighting the difficulty of our proposed dataset. Further, models that performed better on COCO did not consistently perform better on the RF100-VL, indicating that the newer YOLO models might be overfitting to COCO. Lastly, we highlight a discrepancy between reported and reproduced numbers on both COCO and OdinW. Discrepancies in COCO evaluation can be attributed to differences in evaluation toolkits, while discrepancies in ODinW evaluation can be attributed to prior work evaluating models using referential grounding evaluation protocols, while we use standard object detection evaluation protocols. We discuss this further in section 4.1. Following prior work, we use single-class prompts for MLLMs in this table (cf. Appendix A).

| Method | COCO Val | | OdinW-13 | | Roboflow100-VL |
|---|---|---|---|---|---|
| | Reported | Ours | Reported | Ours | Ours |
| **Zero-Shot** | | | | | |
| Qwen 2.5-VL (72B) [28] (Class Names Only) | - | - | 43.1 | 30.9 | 7.8 |
| Gemini 2.5 Pro [47] (Class Names Only) | - | - | 41.9 | 33.7 | 8.0 |
| **Fully-Supervised** | | | | | |
| YOLOv8n [71] | 37.3 | 37.4 | - | - | 54.9 |
| YOLOv11n [74] | 39.5 | 39.4 | - | - | 55.3 |
| YOLOv8s [71] | 44.9 | 45.0 | - | - | 56.2 |
| YOLOv11s [74] | 47.0 | 46.9 | - | - | 56.2 |
| YOLOv8m [71] | 50.2 | 50.3 | - | - | 56.4 |
| YOLOv11m [74] | 51.5 | 51.5 | - | - | 56.5 |

Table 2: **Roboflow100-VL Benchmark.** We evaluate the zero-shot, few-shot, semi-supervised, and fully-supervised performance of state-of-the-art methods on the RF100-VL benchmark. We find that RF100-VL is particularly challenging for zero-shot and few-shot approaches, with most methods struggling to achieves 10% mAP averaged over all 100 datasets. Notably, we find that GroundingDINO achieves the best zero-shot and few-shot accuracy. We use a double horizontal bar to separate specialist models from generalist MLLMs. Note that we use multi-class prompts for MLLMs in this table (cf. Appendix A).

| Method | Aerial | Document | Flora & Fauna | Industrial | Medical | Sports | Other | All |
|---|---|---|---|---|---|---|---|---|
| **Zero-Shot** | | | | | | | | |
| Detic [161] | 12.2 | 4.5 | 17.9 | 6.0 | 0.8 | 7.6 | 11.2 | 9.5 |
| GroundingDINO [86] | 21.5 | 9.2 | 27.9 | 10.3 | 2.1 | 13.3 | 17.5 | 15.7 |
| OWLv2 [97] (Class Names Only) | 19.8 | 12.2 | 23.3 | 7.8 | 2.1 | 12.5 | 14.3 | 13.6 |
| MQ-GLIP-Text [152](Class-Names Only) | 12.1 | 10.0 | 23.2 | 7.8 | 1.4 | 9.3 | 14.2 | 12.2 |
| Qwen 2.5 VL (72B) [28] (Class Names Only) | 4.6 | 3.9 | 10.4 | 4.1 | 1.6 | 6.0 | 5.6 | 5.6 |
| Qwen 2.5 VL (72B) [28] (Instructions Only) | 5.4 | 5.0 | 14.8 | 5.6 | 1.7 | 7.6 | 7.6 | 7.6 |
| Gemini 2.5 Pro [47] (Class Names Only) | 8.7 | 11.8 | 18.3 | 8.6 | 5.3 | 6.5 | 15.4 | 11.6 |
| Gemini 2.5 Pro [47] (Instructions Only) | 1.8 | 6.2 | 7.9 | 3.5 | 0.6 | 2.1 | 5.9 | 4.5 |
| **Few-Shot (10 shots)** | | | | | | | | |
| Detic [161] w/ Federated Loss [91] | 19.5 | 19.6 | 28.4 | 25.9 | 8.5 | 26.6 | 25.7 | 22.8 |
| MQ-GLIP-Image [152] (Images Only) | 4.4 | 3.2 | 13.3 | 3.9 | 1.4 | 7.4 | 6.9 | 6.4 |
| MQ-GLIP [152] (Class Names + Images) | 12.1 | 9.5 | 23.1 | 7.8 | 1.4 | 9.3 | 14.3 | 12.2 |
| GroundingDINO [86] | 32.4 | 30.6 | 41.3 | 37.8 | 18.3 | 33.2 | 32.0 | 33.6 |
| YOLOv8n [71] | 12.8 | 22.8 | 20.9 | 28.1 | 13.7 | 13.6 | 19.9 | 20.2 |
| YOLOv8n [71] w/ Federated Loss [91] | 13.5 | 25.4 | 22.0 | 25.9 | 14.6 | 14.6 | 21.3 | 21.7 |
| YOLOv8s [71] | 15.9 | 22.8 | 22.1 | 24.7 | 13.9 | 18.0 | 21.7 | 20.7 |
| YOLOv8s [71] w/ Federated Loss [91] | 17.4 | 24.9 | 25.4 | 26.5 | 16.8 | 18.5 | 23.3 | 23.6 |
| YOLOv8m [71] | 14.3 | 24.0 | 19.7 | 24.9 | 13.1 | 19.7 | 22.9 | 20.3 |
| YOLOv8m [71] w/ Federated Loss [91] | 16.9 | 23.3 | 20.8 | 26.6 | 16.0 | 21.4 | 22.6 | 22.6 |
| Qwen 2.5 VL (72B)[28] (Instructions + Images) | 5.7 | 6.6 | 14.8 | 5.8 | 1.7 | 7.3 | 6.8 | 7.6 |
| Gemini 2.5 Pro [47] (Images) | 6.2 | 9.4 | 17.5 | 9.5 | 2.7 | 5.0 | 9.7 | 9.8 |
| Gemini 2.5 Pro [47] (Instructions + Images) | 5.3 | 8.8 | 15.0 | 8.8 | 2.1 | 4.9 | 9.5 | 8.8 |

**Do COCO Detectors Generalize Beyond COCO?** Real-time object detectors are often optimized for COCO, assuming better performance on COCO translates to real-world improvements. However, real-world datasets (such as those in RF100-VL) are often much smaller and more diverse than COCO, challenging this assumption. Specifically, although RF100-VL has half as many images as COCO, it has more than seven times as many classes (cf. Fig. 5). Interestingly, we find that models that achieved higher performance on COCO did not necessarily improve real-world performance on RF100-VL. For example, YOLOv11 outperforms YOLOv8 on COCO but performs similarly to YOLOv8 across all three tested sizes (nano, small, medium) on RF100-VL. This suggests that newer YOLO models may be overfitting to COCO, as gains on that dataset don't transfer to real-world datasets. Lastly, we find that increasing model size leads to smaller performance improvements on RF100-VL compared to COCO. The performance difference between the smallest and largest models within a model family is at most 1.9 mAP, suggesting that simply increasing model capacity may not lead to significant performance gains on RF100-VL.

## 4.3 CVPR 2025 Foundational FSOD Challenge

We hosted a challenge at CVPR 2025 to encourage broad community involvement in addressing the problem of aligning foundation models to target concepts with few-shot visual examples and rich textual descriptions. Importantly, we use a subset of 20 datasets from RF100-VL for this challenge to lower the barrier to entry. Our competition received submissions from 25 teams (some submissions are private) at the close of our competition on June 8th, 2025 AOE. Notably, ten teams beat our best baseline. We present the current top three teams in Table 3. We summarize the contributions of the top three teams in Appendix M and include a link to full technical reports and code here.

## 4.4 Limitations and Future Work

**Reliance on Crowdsourced Annotations.** All our datasets are sourced from Roboflow Universe, a community platform where anyone can upload dataset annotations. Although this allows us to source diverse datasets, it introduces uncertainty regarding overall annotation quality. While we manually inspect and re-annotate all datasets to ensure quality to the best of our ability, verifying annotations in specialized domains like medical imaging remains a significant challenge.

Table 3: **CVPR 2025 Foundational FSOD Challenge with Roboflow20-VL.** This year's challenge winner beat our best few-shot baseline by 17 AP! For more details about top performing methods, see Appendix M.

| Method | Aerial | Document | Flora & Fauna | Industrial | Medical | Sports | Other | All |
|---|---|---|---|---|---|---|---|---|
| **Zero-Shot** | | | | | | | | |
| Detic [161] | 4.1 | 1.4 | 22.2 | 6.3 | 0.1 | 1.0 | 9.7 | 8.4 |
| GroundingDINO [86] | 28.5 | 5.1 | 33.7 | 12.8 | 0.4 | 5.1 | 16.9 | 16.8 |
| OWLv2 [97] (Class Names Only) | 35.5 | 4.9 | 24.4 | 12.0 | 0.1 | 3.2 | 12.7 | 14.2 |
| MQ-GLIP-Text [152](Class-Names Only) | 30.1 | 2.5 | 32.8 | 5.5 | 0.5 | 6.4 | 10.8 | 14.0 |
| Qwen 2.5 VL (72B) [28] (Class Names Only) | 3.8 | 3.5 | 10.2 | 2.8 | 0.1 | 9.6 | 3.9 | 5.1 |
| Qwen 2.5 VL (72B) [28] (Instructions Only) | 4.9 | 7.8 | 13.4 | 5.1 | 0.4 | 11.5 | 5.8 | 7.4 |
| Gemini 2.5 Pro [47] (Class Names Only) | 3.5 | 9.7 | 21.5 | 13.3 | 0.4 | 8.9 | 12.2 | 11.5 |
| Gemini 2.5 Pro [47] (Instructions Only) | 1.0 | 8.0 | 9.7 | 7.9 | 0.1 | 4.1 | 4.8 | 5.7 |
| **Few-Shot (10 shots)** | | | | | | | | |
| Detic w/ Federated Loss [91] | 11.6 | 14.3 | 30.8 | 24.7 | 8.9 | 17.4 | 21.0 | 20.3 |
| GroundingDINO [86] | 39.9 | 34.5 | 45.7 | 37.8 | 23.3 | 26.3 | 24.7 | 33.4 |
| MQ-GLIP-Image [152] (Images Only) | 1.8 | 1.1 | 17.6 | 1.8 | 0.1 | 6.6 | 6.8 | 6.7 |
| MQ-GLIP [152] (Class Names + Images) | 29.8 | 2.5 | 32.7 | 5.6 | 0.5 | 6.5 | 10.9 | 14.0 |
| Qwen 2.5 VL (72B) [28] (Instructions + Images) | 5.1 | 9.3 | 15.2 | 2.9 | 0.2 | 8.5 | 5.7 | 7.2 |
| Gemini 2.5 Pro [47] (Images Only) | 7.7 | 14.2 | 24.3 | 4.0 | 0.2 | 12.8 | 9.7 | 9.7 |
| Gemini 2.5 Pro [47] (Instructions + Images) | 8.4 | 12.4 | 12.4 | 19.3 | 0.2 | 8.6 | 4.9 | 8.6 |
| **Challenge Submissions** | | | | | | | | |
| BEATON | 52.4 | 46.9 | 56.3 | 62.0 | 42.9 | 42.0 | 45.8 | 50.4 |
| FDUROILab | 52.3 | 49.9 | 56.9 | 61.6 | 42.1 | 41.9 | 42.4 | 49.8 |
| NJUST-KMG | 49.5 | 43.8 | 57.4 | 59.1 | 42.1 | 42.9 | 43.1 | 49.0 |

**Generated Annotator Instructions May Not Reflect Real Instructions.** Our annotator instructions are automatically generated by GPT-4o and are manually verified for correctness. However, they may not fully reflect the nuances of real-world instructions typically developed alongside dataset collection. We encourage the community to release real annotator instructions generated through iterative discussions between annotators and stakeholders. Furthermore, although our annotator instructions provide high-level class descriptions, they often do not directly incorporate image evidence to identify typical cases, edge cases, and negative examples. Future work should explore how to create better automatic annotator instructions.

**Generalist and Specialist Models have Complementary Strengths.** Although specialist models like GroundingDINO [86] outperform generalist models like Qwen2.5-VL [28], MLLMs can more easily process few-shot visual examples and rich textual descriptions. Future work should combine the versatility of MLLMs with the precision of specialist models.

## 5 Conclusion

In this paper, we introduce Roboflow100-VL, a large-scale benchmark to evaluate state-of-the-art VLMs on concepts not typically found in internet-scale pre-training. RF100-VL is curated to evaluate detection performance on out-of-distribution tasks (e.g. material property estimation, defect detection, and contextual action recognition) and imaging modalities (e.g. X-rays, thermal spectrum data, and aerial imagery) using a few visual examples and rich textual descriptions. We find that state-of-the-art models struggle on this challenging benchmark, demonstrating the limitations of existing methods, highlighting opportunities to develop better algorithms that effectively use multi-modal annotator instructions. We hope that RF100-VL will be a rigorous test-bench for future VLMs and MLLMs.

## 6 Acknowledgments

This work was supported in part by compute provided by NVIDIA, and the NSF GRFP (Grant No. DGE2140739).

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

# A  Implementation Details

We present additional implementation details to reproduce our baseline experiments below. Our code is available on GitHub.

**Detic.** We use Detic [161] with a SWIN-L backbone for all zero-shot experiments. Additionally, we use the model checkpoint trained on LVIS, COCO and ImageNet-21K. We use class names provided as text prompts for Detic's CLIP classifier.

**GroundingDINO.** We use GroundingDINO [86] with pretrained weights from mmdetection (MM-GroundingDINO-L*). We prompt the model with all the class names combined into a single prompt. We fine-tune GroundingDINO on each few-shot dataset for 1000 iterations with a batch size 4 and learning rate of 3e-4. We resize all images to (640, 1333) and don't use any additional data augmentations.

**MQ-GLIP.** MQ-Det [152] proposes a learnable module that enables multi-modal prompting. We choose GLIP with a SWIN-L backbone as the underlying detection model for our experiments. We use the model checkpoint trained on Objects365, FourODs, GoldG, and Cap24M. Laslty, we use class names as the text prompts and few-shot visual examples as visual prompts.

**OWLv2.** We use OWLv2 [97] as implemented in HuggingFace. We prompt the model with each class name independently.

**Qwen-2.5VL.** We conduct all experiments using the "qwen2.5-vl-72b-instruct" model via API. We prompt the model based on guidelines from Qwen's official documentation. We also improve the base prompt through small-scale validation on multiple datasets and select the best prompt:

**System Prompt**

"You are a helpful assistant capable of object detection."

**Multi-Class Detection Prompt**

"Locate all of the following objects: {class names} in the image and output the coordinates in JSON format."

**Single-Class Detection Prompt**

"Locate every {class name} in the image and output the coordinates in JSON format."

**Gemini 2.5 Pro.** We conduct all experiments using the Gemini API with the "gemini-2.5-pro-preview-03-25" model. We prompt the model based on guidelines from Gemini's official documentation, but also improve the base prompt through small-scale validation on multiple datasets and select the best prompt:

**System Prompt**

"Return bounding boxes as a JSON array with labels. Never return masks or code fencing."

**Multi-Class Detection Prompt**

"Detect the 2d bounding boxes of the following objects: {class names}"

**Single-Class Detection Prompt**

> "Detect all 2d bounding boxes of {class name}."

**Prompting with Rich Textual Descriptions** To evaluate Qwen and Gemini with dataset-specific annotator instructions, we appended the following prompt after our main prompt:

> "Use the following annotator instructions to improve detection accuracy: {annotator instructions}"

We include the rich textual description for all classes when using the multi-class detection prompt. In contrast, we only append the relevant class description (extracted using GPT-4o) when using the single-class detection prompt.

**Prompting with Few-Shot Visual Examples** We provide one image at a time to Qwen and Gemini to mimic their turn-based pre-training. We use all few-shot images when prompting Gemini. However, we only use three images when prompting Qwen due to API limitations.

We prompt Gemini with native resolution images, but limit Qwen's few-shot visual examples to a minimum of 4*28*28 pixels and a maximum of 12800*28*28 pixels due to API limitations. To manage costs, we limit Gemini to only output 8192 tokens per request. We do not set any token limits for Qwen. Lastly, we implement a robust parser to handle minor JSON formatting errors. In some cases with many few-shot image examples, the API fails to return a valid response for requests of excessive size. In such cases, we simply assign a score of 0 AP for those images. Due to Gemini and Qwen not always predicting a confidence score for their bounding boxes, we set it to 1.0 by default.

**YOLOv8 and YOLOv11.** We train our YOLOv8 [71] and YOLOv11 [74] family of models using the Ultralytics package with default parameters. For all models, we follow the established protocol in Ciaglia et. al. [38] and train for 100 epochs with a batch size of 16. However, we evaluate all YOLO models using pycocotools instead of Ultralytics (cf. Appendix B)

## B    Additional Evaluation Details

We find that metrics reported with pycocotools (500 maxDets) differs significantly from those reported by Ultralytics on RF100-VL (cf. Table 4). Notably, all YOLO models report metrics using Ultralytics' implementation of mAP by default. Our preliminary investigation, supported by similar observations on Github, suggest that this disparity can be largely attributed to differences in the integration method of the precision-recall curve. Ultralytics uses a trapezoidal sum, which inflates model performance by as much 2.7% compared to pycocotools. We choose to report results for YOLO models using pycocotools in the main paper to standardize our results with our other baselines.

Table 4: **Impact of Evaluation Toolkit on RF100-VL Performance.** We find that the Ultralytics mAP calculation significantly over-estimates mAP compared with pycocotools. For fair comparison with other baselines, we choose to report metrics using pycocotools.

| Method | pycocotools mAP (Ours) | Ultralytics mAP |
|---|---|---|
| YOLOv8n [71] | 55.4 | 57.2 |
| YOLOv11n [74] | 56.1 | 57.8 |
| YOLOv8s [71] | 56.5 | 59.0 |
| YOLOv11s [74] | 57.0 | 59.4 |
| YOLOv8m [71] | 56.9 | 59.6 |
| YOLOv11m [74] | 57.0 | 59.6 |

## C    Ablation on Prompting MLLMs

We evaluate Gemini 2.5 Pro and Qwen 2.5-VL performance on RF100-VL using two prompting strategies: single-class prompting and multi-class prompting. The single-class prompting strategy separately performs a forward pass for each class and merges the results per image. The multi-class prompting strategy performs a single forward pass for all classes. Both Gemini 2.5 Pro and Qwen

2.5-VL recommend the single-class prompting strategy. Importantly, we do not perform non-maximal suppression for either strategy as both MLLMs do not report confidence scores per box.

Interestingly, we observe that Qwen2.5VL performs better with single-class prompts, while Gemini performs better with multi-class prompts. We posit that this can be attributed to Qwen's extensive referential object detection pre-training, which typically requires detecting a single class. In contrast, Gemini achieves better performance with multi-class prompting, which is more aligned with traditional object detection setups. We argue that multi-class prompting should be the default for assessing a MLLM's object detection capabilities since this more closely mirrors standard object detection protocols.

Table 5: **Analysis of Prompting Strategy.** MLLMs typically evaluate detection performance with single-class prompts. We find that Qwen2.5VL achieves better performance with single-class prompts, while Gemini 2.5 Pro achieves better performance with multi-class prompts. We advocate for multi-class prompting since this more closely matches object detection evaluation.

| Method | Single-Class Prompt | Multi-Class Prompt |
|---|---|---|
| Gemini 2.5 Pro | 8.0 | 11.6 |
| Qwen 2.5-VL (72B) | 7.8 | 5.6 |

# D   Comparing Different Model Sizes

In Figure 6, we evaluate the performance of the Gemini model family over time (e.g. Gemini Flash 2.0 was released before Gemini Flash 2.5). Although Gemini has not been explicitly fine-tuned on RF100-VL, we see a significant increase in performance. This suggests that Gemini is making real progress towards zero-shot open-vocabulary object detection in the wild. Unsurprisingly, base MLLMs outperform faster distilled models (e.g. Gemini 2.5 Pro achieves 35% better performance than Gemini Flash 2.5), but distilled models provide considerably better performance per dollar. Importantly, all models are prompted with multi-class prompts (cf. Appendix C).

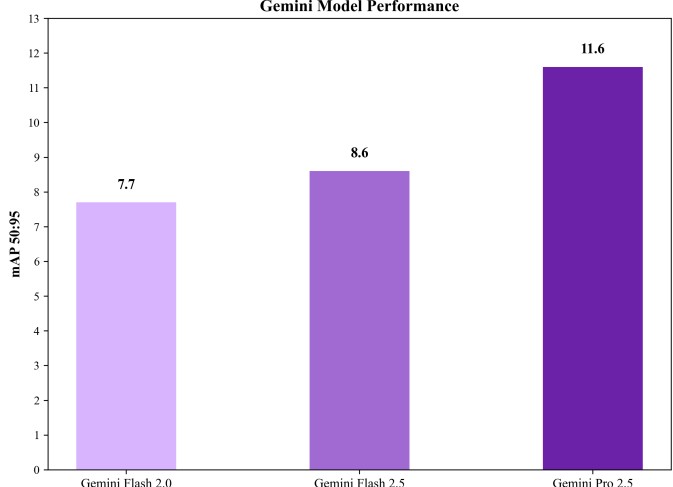

Figure 6: **Gemini Improves on RF100-VL over Time**. Despite not explicitly fine-tuning on RF100-VL, we find that newer Gemini models consistently improve over older models on our benchmark. This suggests that Gemini is making real progress towards improving zero-shot open-vocabulary detection in-the-wild.

# E   Ablation on Few-Shot Split Selection

Prior work typically selects few-shot training examples at random. However, Madan et. al. [91] demonstrates that the specific few-shot examples used for fine-tuning greatly affects target class performance. Specifically, Madan et. al. selects the most informative K-shot examples for each class in nuImages [32] by evaluating Detic w/ Federated Fine-Tuning's class-wise performance on a held-out validation set. For instance, in a 5-shot task with three random splits, we may select our five-shot car examples from split 1, our five-shot bicycles from split 3, and our five-shot debris from

split 2 based on which split has the highest per-class accuracy. As shown in Table 6, this "best split [91]" approach consistently outperforms random selection.

Despite the effectiveness of this approach, it has two primary limitations. First, it uses the validation performance of a specific model to inform few-shot selection. This inherently biases the few-shot images towards a particular model. Next, Madan et. al.'s proposed algorithm is computationally expensive since it requires fine-tuning a model on many candidate few-shot splits. This approach is computationally infeasible with RF100-VL's 100 datasets.

To address these two issues, we propose a learning-free approach that leverages the key insight from Madan et. al's analysis: the "best" examples are typically large and unoccluded. Concretely, we generate $K$ random candidate few-shot splits for each class and pick the split that has the largest average bounding box area. Similar to Madan et. al., in a 5-shot task with three random splits, we may select our five-shot car examples from split 1, our five-shot bicycles from split 3, and our five-shot debris from split 2. We evaluate our proposed sampling strategy on nuImages and find that this approach performs better than random, but underperforms Madan et. al.'s approach. Future work should consider more effective strategies for selecting the "best" few-shot examples for concept alignment.

Table 6: **"Best" Split Construction**. We evaluate the quality of few-shot example selection using a (1) random baseline, (2) Madan et al.'s "best split" approach, which chooses per-class few-shot examples based on Detic w/ Federated Fine-Tuning's validation accuracy, and (3) our proposed learning-free method that selects splits with the largest average bounding box area. While Madan et al.'s method performs the best, it is biased towards Detic and is computationally expensive. Our approach offers a tractable alternative that improves over the random baseline.

| Approach | Average Precision (AP) | | | |
| --- | --- | --- | --- | --- |
| | All | Many | Medium | Few |
| Detic (Zero-Shot) [161] | 14.40 | 25.83 | 16.59 | 2.32 |
| Detic w/ Federated Fine-Tuning *(5-shots, Random Split)* | 16.58 | 27.12 | 19.71 | 4.13 |
| Detic w/ Federated Fine-Tuning *(5-shots, Best Split [91])* | **18.30** | **28.66** | **21.81** | **5.56** |
| Detic w/ Federated Fine-Tuning *(5-shots, Best Split, Ours)* | 16.94 | 28.41 | 20.32 | 3.45 |
| Detic w/ Federated Fine-Tuning *(10-shots, Random Split)* | 17.24 | 28.07 | 20.71 | 4.18 |
| Detic w/ Federated Fine-Tuning *(10-shots, Best Split [91])* | **18.24** | **28.63** | 22.00 | **5.19** |
| Detic w/ Federated Fine-Tuning *(10-shots, Best Split, Ours)* | 17.48 | 26.36 | **22.42** | 4.32 |

## F  Semi-Supervised and Fully Supervised Results

We present results from semi-supervised and fully-supervised baselines in Table 7. Importantly, these models are evaluated on the same data splits as our zero-shot and few-shot baselines. To construct the semi-supervised split, we randomly sample 10% of the training set.

**Semi-Supervised Baselines.** We evaluate variants of YOLO [71, 74] and YOLO with STAC [127] trained on 10% of each dataset in RF100-VL. STAC generates high-confidence pseudo-labels for localized objects in unlabeled images and updates the model by enforcing consistency through strong augmentations. We follow the training protocol defined by Sohn et. al. [127]. First, we train a teacher model on the labeled subset of the data. Then, we use the teacher model to pseudo-label the remaining unlabeled subset of the data. We keep all detections above a confidence $C$, where the confidence tuned to maximize the F1 score of the teacher model on a validation set. Finally, we combine the subset of data with true ground truth labels and the subset with pseudo-labels to form a training set for a student model of the same architecture. We train this student model until convergence with heavy augmentations. We use the same hyperparameters as our supervised YOLOv8 and YOLOv11 implementation. Because YOLO models already train with significant augmentation, we don't add any new augmentations for the student training.

**Fully-Supervised Baselines.** We benchmark YOLOv8 [71], YOLOv11 [74], and LW-DETR [36] on all datasets within RF100-VL. YOLOv8, developed by Ultralytics, builds on the YOLOv5 architecture with improvements in model scaling and architectural refinements. YOLOv11 adds more architecture improvements, and is primarily validated on COCO. LW-DETR is a lightweight detection transformer that outperforms YOLO models for real-time object detection, and is SOTA on the original Roboflow100 [38] dataset, the predecessor to RF100-VL. Its architecture consists of a ViT encoder, a projector, and a shallow DETR decoder. This baseline serves as an upper bound on

performance, though in rare cases, few-shot foundation models may surpass it when the target dataset only has a few examples.

**Semi-Supervised Learners are Data Efficient.** We find that leveraging simple semi-supervised learning algorithms like STAC [127] significantly improves model performance when learning with limited labels. In half (7 out of 14) of combinations of model size and data domain, semi-supervised learners improved mAP at least as much as stepping up a model size. For example, YOLOv8s (small) trained on 10% labeled data (and 90% STAC psuedo-labels) achieves better performance overall than YOLOv8m (medium) trained on just 10% labeled data.

Table 7: **Roboflow100-VL Semi-Supervised and Fully-Supervised Benchmark.** We find that semi-supervised learners are able to reach nearly 80% of the performance of fully supervised models using 10% labeled data.

| Method | Aerial | Document | Flora & Fauna | Industrial | Medical | Sports | Other | All |
|---|---|---|---|---|---|---|---|---|
| **Semi-Supervised (10% Labels)** | | | | | | | | |
| YOLOv8n [71] | 32.8 | 35.1 | 41.5 | 50.7 | 30.2 | 29.4 | 37.6 | 39.3 |
| YOLOv8n [71] w/ STAC [127] | 35.3 | 38.5 | 43.6 | 51.5 | 31.5 | 32.1 | 40.1 | 41.2 |
| YOLOv8s [71] | 37.8 | 40.8 | 42.6 | 52.6 | 32.8 | 36.8 | 41.3 | 42.3 |
| YOLOv8s [71] w/ STAC [127] | 38.2 | 42.7 | 43.4 | 52.4 | 34.0 | 38.4 | 42.3 | 43.1 |
| YOLOv8m [71] | 37.8 | 40.5 | 41.3 | 52.9 | 33.4 | 41.4 | 42.1 | 42.5 |
| YOLOv8m [71] w/ STAC [127] | 37.5 | 42.5 | 43.2 | 52.7 | 34.2 | 40.2 | 43.5 | 43.3 |
| **Fully-Supervised** | | | | | | | | |
| YOLOv8n [71] | 50.4 | 56.4 | 53.9 | 64.3 | 50.0 | 49.2 | 54.6 | 55.4 |
| YOLOv11n [74] | 51.2 | 58.4 | 54.8 | 64.6 | 50.3 | 49.2 | 55.5 | 56.1 |
| YOLOv8s [71] | 51.6 | 58.9 | 54.9 | 64.6 | 50.2 | 51.2 | 56.7 | 56.5 |
| YOLOv11s [74] | 53.1 | 58.8 | 55.5 | 64.7 | 50.3 | 52.0 | 57.4 | 57.0 |
| LW-DETRs [36] | 54.5 | 57.7 | 54.2 | 66.8 | 51.7 | 54.7 | 56.3 | 57.4 |
| YOLOv8m [71] | 52.5 | 59.9 | 55.1 | 64.7 | 49.5 | 52.6 | 57.5 | 56.9 |
| YOLOv11m [74] | 53.0 | 60.5 | 54.8 | 65.1 | 49.9 | 52.4 | 57.3 | 57.0 |
| LW-DETRm [36] | 57.1 | 60.1 | 56.7 | 68.2 | 52.8 | 57.5 | 61.0 | 59.8 |

# G  Analysis of Accuracy vs. Parameter Count

In Figure 7, we observe a counter-intuitive trend: larger models perform worse in our evaluations. This is likely due to the mismatch between general-purpose MMLMs and specialized object detectors. Despite being the largest model pre-trained on the most data, Qwen2.5-VL (72B) underperforms GroundingDINO in the zero-shot setting and is also considerably slower. Interstingly, we find that GroundingDINO fine-tuned on few-shot examples surpasses all YOLO models fine-tuned on few-shot examples, indicating that large pre-trained backbones enable more efficient fine-tuning in specialist models.

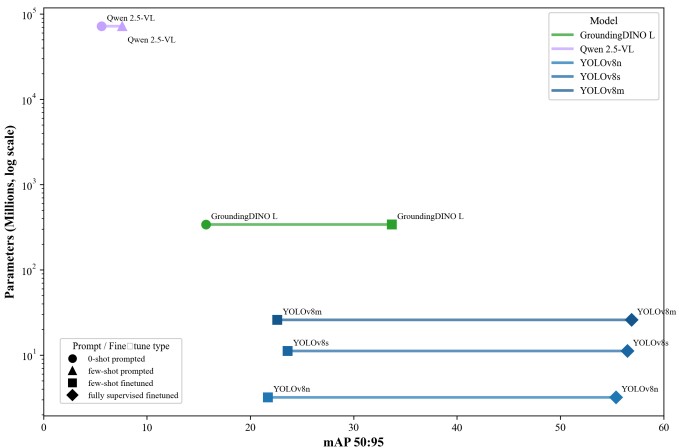

Figure 7: **Accuracy vs. Parameter Count.** Somewhat counterintuitively, we find that the model with the most parameters (Qwen2.5-VL 72B) performs worse than significantly smaller models pre-trained on less data (GroundingDINO) in the zero-shot setting. This suggests that generalist MLLMs are parameter inefficient for specialized tasks.

# H  Correlation Between Model Type and Per-Dataset Performance

Figure 8 presents four scatterplots comparing mAP 50:95 across different model pairs on RF100-VL, with each axis representing one model's mAP and each point labeled by a dataset index (sorted

alphabetically). These plots help identify whether certain datasets are universally easy, medium, or hard across models.

We compare Gemini vs. GroundingDINO, Qwen vs. GroundingDINO, Gemini vs. Qwen, and GroundingDINO vs. YOLO. Gemini and Qwen, as well as GroundingDINO and YOLO, show stronger linear correlations in their per-dataset scores, suggesting alignment in perceived difficulty. In contrast, comparisons between generalists (Gemini and Qwen) and specialists (GroundingDINO and YOLO) show weaker correlation. This suggests that large-scale MLLMs, likely trained on similar web data, align more closely with each other, while specialist models like GroundingDINO and YOLO show stronger consistency. These results imply that dataset difficulty levels (easy, medium, hard) may not generalize across model classes, but may be better defined within model types.

Additionally, among the top 15 datasets where Gemini outperforms Qwen and GroundingDINO, seven overlap. This suggests that Gemini may excel on datasets similar to those found in its pretraining, but struggles to generalize to novel domains.

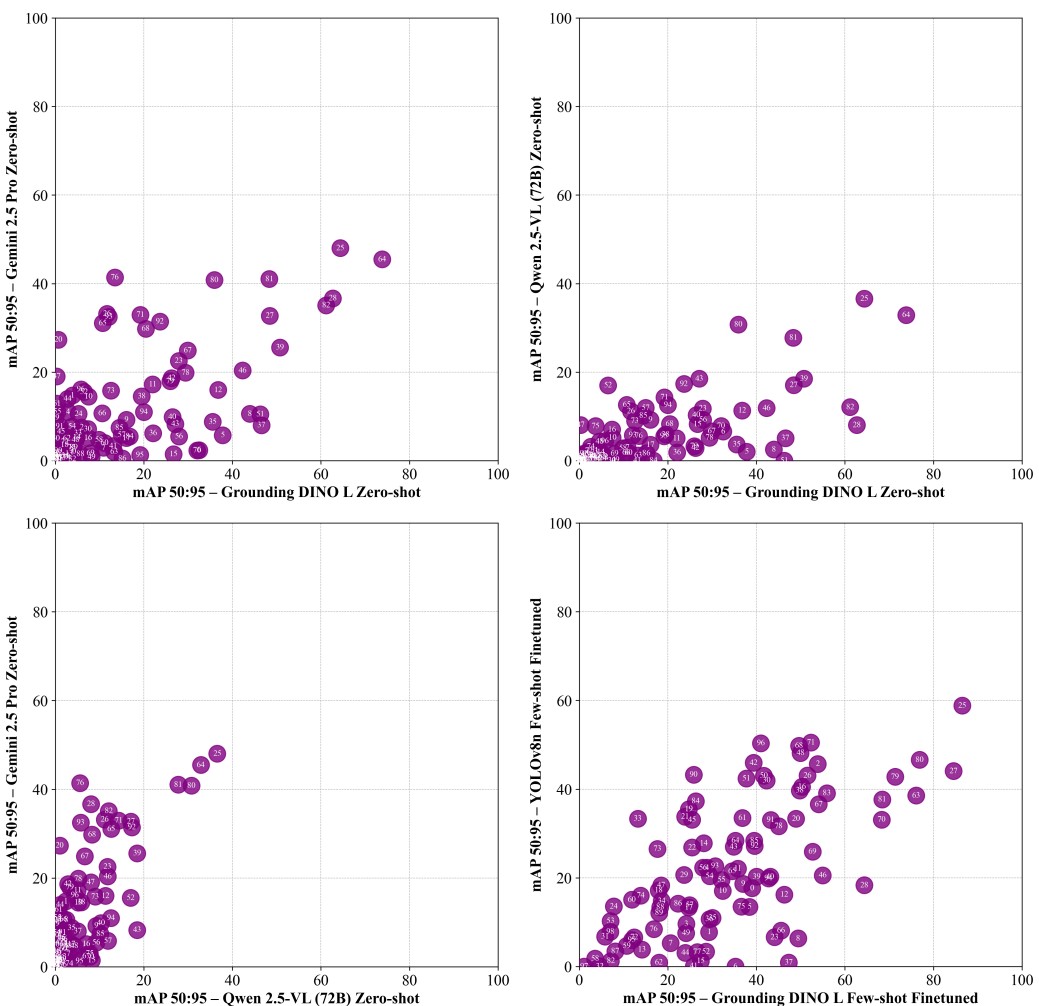

Figure 8: **Correlation Between Models Type and Performance.** We see stronger linear trends between Gemini and Qwen, and between GroundingDINO and YOLO, indicating aligned perceptions of dataset difficulty within model groups.

# I  Performance Variance for Few-Shot Models

Tables 8 and 9 measure the variance of YOLOv8 on RF100-VL. We use this model as a proxy for understanding few-shot learning variance and the statistical significance of our results. We train

YOLOv8n and YOLOv8s [71] with federated loss [91] ten times on each dataset, using ten different random seeds to determine model initialization and augmentation selection. We report the mean and standard deviation in two ways. In Table 8, we take the average mAP across all datasets in a given category (e.g. Industrial, Sports, All, etc.), and report the mean mAP and standard deviation across ten different runs. In Table 9, we measure the mean and standard deviation for each dataset across 10 different runs, and then report the average mean and standard deviation over each category. This will result in a higher standard deviation. Table 9 conveys the variance of a single dataset in RF100-VL and motivates averaging mAP across multiple datasets as a more stable metric.

Table 8: **Roboflow100-VL Overall Variance.** We evaluate the mean mAP and standard deviation of ten runs of YOLOv8 [71] with federated loss [91] over different subsets of Roboflow100-VL. These results can be used as a proxy to calculate whether a new entry to Table 2 is statistically significant. Unsurprisingly, averaging over 100 datasets yields a less noisy estimate of model performance

| Method | Aerial | Document | Flora & Fauna | Industrial | Medical | Sports | Other | All |
|---|---|---|---|---|---|---|---|---|
| YOLOv8n [71] | $13.5 \pm .778$ | $24.9 \pm 1.19$ | $20.8 \pm .246$ | $30.1 \pm .409$ | $15.9 \pm .771$ | $14.8 \pm 1.270$ | $21.8 \pm .891$ | $21.6 \pm .230$ |
| YOLOv8s [71] | $17.3 \pm .791$ | $26.4 \pm .902$ | $23.4 \pm .407$ | $30.0 \pm .680$ | $17.8 \pm .565$ | $19.2 \pm .536$ | $25.0 \pm .252$ | $23.7 \pm .216$ |

Table 9: **Roboflow100-VL Dataset Variance.** We evaluate the mean mAP and standard deviation over 10 runs of YOLOv8 [71] with federated loss [91] for each of the 100 datasets in Roboflow100-VL. These results helps quantify how much a model should improve on a single dataset to be statistically significant. This approach for quantifying statistical significance shows a much higher variance.

| Method | Aerial | Document | Flora & Fauna | Industrial | Medical | Sports | Other | All |
|---|---|---|---|---|---|---|---|---|
| YOLOv8n [71] | $13.5 \pm 2.29$ | $24.9 \pm 2.65$ | $20.8 \pm 2.80$ | $30.1 \pm 2.48$ | $15.9 \pm 2.21$ | $14.8 \pm 2.07$ | $21.8 \pm 2.55$ | $21.6 \pm 2.50$ |
| YOLOv8s [71] | $17.3 \pm 2.25$ | $26.4 \pm 2.86$ | $23.4 \pm 3.24$ | $30.0 \pm 2.88$ | $17.8 \pm 2.43$ | $19.2 \pm 1.92$ | $25.0 \pm 2.44$ | $23.7 \pm 2.71$ |

# J   Impact of Instruction Quality

We evaluate few-shot detection performance on RF20-VL using annotator instructions generated by GPT4o, Qwen 2.5-VL, Gemini 2.5 Pro, GPT4o with a human-in-the loop (our original instructions), and human written instructions in Table 10. We evaluate the impact of instruction source on Qwen 2.5 VL and Gemini 2.5 Pro. Notably, we do not find a clear correlation between instruction source and downstream model performance. We find that Qwen 2.5 VL achieves better performance with annotator instructions from all sources compared to class names only, while Gemini 2.5 Pro performs worse with annotator instructions from all sources compared to class names only. Somewhat surprisingly, we find that instructions from GPT 4o with a human-in-the-loop performs the best on both Qwen 2.5VL and Gemini 2.5 Pro, beating human written instructions. Although prompting with annotator instructions yields inconsistent benefits, future work should explore novel ways of incorporating such rich contextual information.

Table 10: **Impact of Instruction Origin.** We find that there is no strong correlation between instruction origin and MLLM detection accuracy.

| Instruction Source | Qwen 2.5VL | Gemini 2.5 Pro |
|---|---|---|
| Class Names Only | 5.1 | 11.5 |
| GPT4o Instructions | 6.4 | 5.3 |
| Qwen 2.5 VL Instructions | 6.4 | 4.7 |
| Gemini 2.5 Pro Instructions | 7.2 | 5.2 |
| GPT-4o Instructions with Edits (Main Paper) | 7.4 | 5.7 |
| Human Written Instructions | 6.6 | 4.4 |

On average, the class names only prompts had 31.75 words, the GPT4o instructions with a human-in-the-loop had 502 words, the shortened GPT4o instructions with a human-in-the-loop had 170.55 words, and the human instructions had 482.95 words. Somewhat surprisingly, we find that the length of the prompt does not correlate well with model performance (cf. Table 11). This suggests that the models are not fine-tuned to leverage such instructions, regardless of context length. Future work should consider more adaptive prompt designs or fine-tuning strategies.

Table 11: **Impact of Instruction Length.** We find that there is no strong correlation between instruction length and MLLM detection accuracy.

| Instruction Source | Qwen 2.5VL | Gemini 2.5 Pro |
|---|---|---|
| Class Names Only | 5.1 | 11.5 |
| GPT-4o Instructions with Edits (Main Paper) | 7.4 | 5.7 |
| Shortened GPT-4o Instructions with Edits | 5.9 | 5.4 |
| Human Instructions | 6.6 | 4.4 |

## K   Impact of Detector-Style Post-Processing with MLLMs

Unlike specialist detectors, MLLMs directly predict bounding boxes without confidence scores, and do not leverage common detector post-processing techniques like NMS. We investigate the impact of such post-processing steps on MLLM detection accuracy with RF20-VL in Table 12. First, we estimate the confidence score of each predicted bounding box with SigLIPv2 [136]. We compute the cosine similarity of the predicted class name text embedding and bounding box image crop embedding. Although one can prompt an MLLM to predict its own confidence scores in theory, recent work [153] demonstrates that MLLMs struggle to verbalize confidence estimates in practice. We expect that the challenging out-of-distribution classes in RF20-VL make directly verbalizing confidence estimates even more difficult. Next, we run NMS per-class. Importantly, these post-processing steps can be applied to both zero-shot and few-shot prompted MLLMs. We find that adding confidence scores from SigLIP significantly improves performance for both Qwen 2.5VL and Gemini 2.5 Pro. Further, NMS seems to have a negligible impact, suggesting the LLMs implicitly learn to avoid making duplicate predictions.

Table 12: **Impact of Detector-Style Post Processing.** We find that adding confidence scores from SigLIP significantly improves performance for both Qwen 2.5VL and Gemini 2.5 Pro, but NMS seems to have negligible impact.

| Model | Qwen 2.5VL | Gemini 2.5 Pro |
|---|---|---|
| **Instructions** | 7.4 | 5.7 |
| + SigLIPv2 Score | 9.7 | 8.3 |
| + NMS | 9.8 | 8.4 |
| **Instructions + Images** | 7.2 | 8.6 |
| + SigLIPv2 Score | 8.8 | 10.6 |
| + NMS | 8.9 | 10.6 |

## L   Analysis of Failure Cases

We can infer the causes of model failure by comparing the relative performance of standard object detectors (e.g. YOLOv8) with VLMs (e.g. Detic) for few-shot object detection. Importantly, unlike Detic, YOLOv8 is not pre-trained on large-scale datasets and is not promptable with class names. Therefore, when both models achieve low performance, we can attribute model failures to difficulties in feature extraction. In contrast, when YOLOv8 performs well, but Detic performance suffers, we can attribute model failures to semantic ambiguity.

Using this heuristic, we analyze YOLOv8m w/ Federated Loss and Detic w/ Federated Loss because they have similar overall performance on RF100-VL (cf. Table 2). Notably, we find that Detic achieves 19.6 AP on Documents, while YOLOv8m achieves 23.3 AP. This suggests that these datasets contain many semantically ambiguous classes. Similarly, Detic achieves 8.5 AP on Medical while YOLOv8m achieves 16.0 AP. In contrast, we posit that datasets where Detic outperforms YOLOv8 (like Flora & Fauna and Sports) are semantically unambiguous and more similar to Detic's pre-training.

## M   Summary of CVPR 2025 Competition Top Performers

We summarize the contributions of top teams below. We present full technical reports and code here.

**BEATON** uses Nebula-CV as the base detector, an unpublished model built on the DINO architecture with Swin-B as the visual backbone and BERT as the text encoder, enabling open-set detection through cross-modal fusion. The model is pre-trained in two stages: first on five million curated

web-scale images, then fine-tuned on one million high-quality grounding examples distilled from Qwen2.5-VL. To address the few-shot setting, they introduce strategies including optimized text prompts generated with Qwen2.5-VL, a carefully tuned combination of data augmentations (e.g., flip, crop, HSV augmentation, copy-paste), pseudo-labeling to supplement sparse annotations, and dataset-specific inference resolution selection. They also tested but found minimal benefit from federated fine-tuning and LLM-based post-processing.

**FDUROILab** proposes a structured fine-tuning strategy enhanced by aggressive data augmentation techniques such as CachedMosaic, YOLOXHSVRandomAug, CachedMixUp, and RandomCrop, which increase data diversity and model robustness. The team employs MM-GroundingDINO with a Swin-L backbone as the base detector and uses Qwen2.5-VL-32B for post-processing to refine classification results by correcting errors made by the primary detector. Training is conducted across 20 datasets, each undergoing 50 independent runs to ensure robust optimization. Their ablation study shows a stepwise improvement in performance, with significant gains from fine-tuning, additional augmentations, multiple training runs, and the MLLM-based post-processing.

**NJUST-KMG** integrates dynamic data augmentation, feature consistency regularization, a dynamic freezing mechanism, grid search optimization, and inference enhancements via Test-Time Augmentation (TTA) and Weighted Boxes Fusion (WBF). The augmentation pipeline dynamically adjusts the probabilities of CachedMosaic, MixUp, HSV jitter, and RandomCrop based on training progression, while the freezing strategy customizes parameter updates depending on dataset size and domain similarity. NJUST-KMG also uses a grid search process to tune hyperparameters and configurations for each dataset to maximize validation mAP. During inference, predictions are refined by combining outputs from the top models using WBF with confidence calibration.

## N  Dataset Comparison

We present a detailed comparison of RF100-VL with related datasets in Table 13. Notably, RF100-VL is the only dataset to support rich textual descriptions and evaluates models in the zero-shot, few-shot, semi-supervised, and fully-supervised data regimes.

Table 13: **Dataset Comparison**. We compare the characteristics of RF100-VL with Roboflow100, COCO, LVIS, Objects365, and OdinW. Notably, RF100-VL is the only dataset with rich textual descriptions and evaluates models across data regimes.

| Dataset | # Datasets | # Images | # Annotations | # Classes | Rich Textual Descriptions | Multi-Spectral Imagery | Zero-Shot Evaluation | Few-Shot Evaluation | Semi-Supervised Evaluation | Fully-Supervised Evaluation | Image Source |
|---|---|---|---|---|---|---|---|---|---|---|---|
| Roboflow100-VL | 100 | 164,149 | 1,355,491 | 564 | ✓ | ✓ | ✓ | ✓ | ✓ | ✓ | Roboflow Universe |
| Roboflow100 [38] | 100 | 224,714 | 1,319,307 | 805 | ✗ | ✓ | ✗ | ✗ | ✗ | ✓ | Roboflow Universe |
| COCO [85] | 1 | 328,000 | 2,500,000 | 91 | ✗ | ✗ | ✗ | ✗ | ✗ | ✓ | Flickr |
| LVIS (v0.5) [65] | 1 | 82,000 | 745,000 | 1230 | ✗ | ✗ | ✗ | ✗ | ✗ | ✓ | COCO |
| Objects365 [124] | 1 | 638,000 | 10,101,000 | 365 | ✗ | ✗ | ✗ | ✗ | ✗ | ✓ | Flickr |
| OdinW [82] | 35 | 152,384 | 1,073,455 | 314 | ✗ | ✗ | ✓ | ✓ | ✗ | ✓ | Roboflow Universe |

## O  RF100-VL Bounding Box Annotation Refinement

We hired external contractors to refine RF100-VL's bounding box annotations according to a set of guidelines, described below. In total, 30 annotators spent 2168 hours validating annotations, with the authors performing additional quality control. Our annotation guidelines provide instructions for improving the quality of existing annotations across datasets. Our primary goal was to ensure consistent annotation style, with every possible instance of each class labeled by a single, tightly fitting bounding box.

The annotation refinement process emphasizes the following corrections:

- *Merged Bounding Boxes*: Annotators must ensure each object has its own bounding box, redrawing boxes that encompass multiple instances of an object.

- *Incomplete Bounding Boxes*: Bounding boxes must fully contain the entire object. If an object extends beyond the box's boundaries, the box needs to be expanded to include the whole object.

- *Missing Annotations*: It is crucial to identify and label every instance of each class within a dataset. This is the most challenging and important aspect of refinement. Annotators are advised to check the background for missing annotations.
- *Incorrectly Labeled Objects*: Bounding boxes around objects that do not belong to the specified class must be removed.
- *Wrong Class Names*: Annotators need to correct instances where class names are misassigned to objects (e.g., doors labeled as windows, or generic numerical labels instead of descriptive class names like "enemy" or "head").
- *Duplicated Bounding Boxes*: If the same object has multiple bounding boxes, one of the duplicates must be removed.

These instructions focus on correcting annotations within each dataset to achieve consistency. When inconsistencies are found in a dataset's labeling scheme, annotators are instructed to make a determination for consistency and ensure all images follow that scheme.

# P    Annotation Generation Instructions

We present our prompt for generating multi-modal annotator with GPT-4o below.

```
Pay attention to the following example annotation instructions for nu-images,
an object detection dataset:

{nuImages Annotator Instructions}

That was an example of object detection annotation instructions.

Using the above instructions as rough inspiration, come up with annotation
instructions for a dataset.

The annotation instructions should be in markdown format, and follow the
following outline:

'''markdown
# Overview
Table of contents

# Introduction
Introduction to the dataset. Introduce what task the dataset is trying to
solve. List all of the classes and provide a brief description of each class.

# Object Classes
## Class 1
### Description
Provide a description of the class, paying attention to visually distinctive
elements of the class.
### Instructions
Provide detailed instructions for how to annotate this class. Give specific
references to the class, and pay attention to the example labeled images that
will be provided. Provide specific descriptions of what not to label, if applicable.
...

## Class 2
...

## Class n
...
'''
```

Please pay specific attention to the provided visual example images and ground
your response in those examples. Be brief and concise, but comprehensive.
Make sure ### Instructions in each class provides visual descriptions of what
exactly to annotate.

Visual descriptions should make specific reference to how the object looks in
each image. If the object is not something everyone knows, describe its distinctive
shape, color, texture, etc. Look at the example pictures when coming up with these
instructions.

Respond with only the markdown content, no other text (and no backticks). Do not
describe the color of the bounding box, just describe how to find the spatial extent
of the object in the image.

The final markdown file should not make specific reference to the provided example
images. Those are simply to help you come up with the instructions. An annotator
should be able to recreate the annotations in the example images using your
generated instructions.

If the classes are similar, make sure the instructions specify how to disambiguate
between them (visually, which specific visual features to look for).

The visual content of the image should be used to clarify the description of each
class. Feel free to generalize about what is present in the dataset from the example
images.

Here is general metadata about the dataset:
{Metadata}

Here are the class names:
{Class Names}

Here are the example images:
{Few-Shot Example Images}

# Q    Sample Annotation Instructions

We present sample annotator instructions below. We use dataset metadata, class names and few-shot
visual examples and prompt GPT-4o [24] to generate annotator instructions (cf. Appendix P). We
then manually verify that the instructions accurately describe the few-shot examples. These annotator
instructions are from recode-waste-czvmg-fsod-yxsw.

# Overview
- [Introduction](#introduction)
- [Object Classes](#object-classes)
    - [Aggregate](#aggregate)
    - [Cardboard](#cardboard)
    - [Hard Plastic](#hard-plastic)
    - [Metal](#metal)
    - [Soft Plastic](#soft-plastic)
    - [Timber](#timber)

# Introduction
This dataset is designed for waste classification within different material
classes. The goal is to accurately identify and annotate different types of
waste materials for sorting and recycling purposes. The classes represented
are: Aggregate, Cardboard, Hard Plastic, Metal, Soft Plastic, and Timber.

# Object Classes

## Aggregate
### Description
Aggregate refers to small, granular materials, often irregular in shape with rough surfaces. They generally appear as pieces of stone or concrete.

### Instructions
Annotate all visible portions of aggregate items. Ensure to include entire objects even if occluded by other materials, estimating boundaries if necessary. Exclude dust or very fine particles that do not form distinct objects.

## Cardboard
### Description
Cardboard objects are typically flat and have a layered texture. They may appear as boxes or sheets.

### Instructions
Annotate only distinguishable pieces of cardboard, focusing on their flat surfaces and any visible layering. Do not annotate cardboard that is part of another object or soiled beyond recognition.

## Hard Plastic
### Description
Hard plastics are rigid and maintain their shape. They can be cylindrical, tubular, or robust objects often found in industrial contexts.

### Instructions
Annotate the entire visible area of hard plastic objects, ensuring to capture their solid structure. Avoid labeling small, indistinct pieces or any plastic that appears flexible.

## Metal
### Description
Metal objects are robust, often shiny or reflective. They can appear as rods, sheets, or other distinct shapes.

### Instructions
Label all distinct metal parts, taking care to capture their complete form. Avoid labeling rust marks or indistinct metallic fragments lacking shape.

## Soft Plastic
### Description
Soft plastics are flexible and often transparent or translucent. They may appear in the form of bags or wrappers.

### Instructions
Focus on full pieces of soft plastic material, ensuring to include areas with visible creases or folds indicating flexibility. Do not label pieces smaller than a recognizable package or those mixed with other materials.

## Timber
### Description
Timber objects are wooden, either rough or smooth, often elongated or rectangular.

### Instructions
Annotate the entire visible portion of timber, focusing on the grain or wood texture. Do not label splinters or fragments that do not exhibit a clear wooden structure.

# R  Roboflow100-VL Datasets

We present a table with links to all datasets within Roboflow100-VL (fully-supervised and FSOD datasets) below.

| Flora & Fauna | Link |
|---|---|
| aquarium-combined [116] | FSOD, Fully Supervised |
| bees [3] | FSOD, Fully Supervised |
| deepfruits [119] | FSOD, Fully Supervised |
| exploratorium-daphnia [30] | FSOD, Fully Supervised |
| grapes-5 [23] | FSOD, Fully Supervised |
| grass-weeds [7] | FSOD, Fully Supervised |
| gwhd2021 [64] | FSOD, Fully Supervised |
| into-the-vale [104] | FSOD, Fully Supervised |
| jellyfish [57] | FSOD, Fully Supervised |
| marine-sharks [44] | FSOD, Fully Supervised |
| orgharvest [150] | FSOD, Fully Supervised |
| peixos-fish [10] | FSOD, Fully Supervised |
| penguin-finder-seg [107] | FSOD, Fully Supervised |
| pig-detection [54] | FSOD, Fully Supervised |
| roboflow-trained-dataset [120] | FSOD, Fully Supervised |
| sea-cucumbers-new-tiles [43] | FSOD, Fully Supervised |
| thermal-cheetah [14] | FSOD, Fully Supervised |
| tomatoes-2 [94] | FSOD, Fully Supervised |
| trail-camera [15] | FSOD, Fully Supervised |
| underwater-objects [17] | FSOD, Fully Supervised |
| varroa-mites-detection–test-set [29] | FSOD, Fully Supervised |
| wb-prova [140] | FSOD, Fully Supervised |
| weeds4 [147] | FSOD, Fully Supervised |

| Industrial | Link |
|---|---|
| -grccs [141] | FSOD, Fully Supervised |
| 13-lkc01 [26] | FSOD, Fully Supervised |
| 2024-frc [22] | FSOD, Fully Supervised |
| aircraft-turnaround-dataset [25] | FSOD, Fully Supervised |
| asphaltdistressdetection [50] | FSOD, Fully Supervised |
| cable-damage [4] | FSOD, Fully Supervised |
| conveyor-t-shirts [49] | FSOD, Fully Supervised |
| dataconvert [137] | FSOD, Fully Supervised |
| deeppcb [148] | FSOD, Fully Supervised |
| defect-detection [42] | FSOD, Fully Supervised |
| fruitjes [70] | FSOD, Fully Supervised |
| infraredimageofpowerequipment [142] | FSOD, Fully Supervised |
| ism-band-packet-detection [39] | FSOD, Fully Supervised |
| l10ul502 [126] | FSOD, Fully Supervised |
| needle-base-tip-min-max [138] | FSOD, Fully Supervised |
| recode-waste [20] | FSOD, Fully Supervised |
| screwdetectclassification [40] | FSOD, Fully Supervised |
| smd-components [100] | FSOD, Fully Supervised |
| truck-movement [16] | FSOD, Fully Supervised |
| tube [51] | FSOD, Fully Supervised |
| water-meter [33] | FSOD, Fully Supervised |
| wheel-defect-detection [55] | FSOD, Fully Supervised |

| Document | Link |
|---|---|
| activity-diagrams [1] | FSOD, Fully Supervised |
| all-elements [66] | FSOD, Fully Supervised |
| circuit-voltages [5] | FSOD, Fully Supervised |
| invoice-processing [143] | FSOD, Fully Supervised |
| label-printing-defect-version-2 [93] | FSOD, Fully Supervised |
| macro-segmentation [45] | FSOD, Fully Supervised |
| paper-parts [9] | FSOD, Fully Supervised |
| signatures [11] | FSOD, Fully Supervised |
| speech-bubbles-detection [21] | FSOD, Fully Supervised |
| wine-labels [18] | FSOD, Fully Supervised |

| Medical | Link |
|---|---|
| canalstenosis [58] | FSOD, Fully Supervised |
| crystal-clean-brain-tumors-mri-dataset [46] | FSOD, Fully Supervised |
| dentalai [19] | FSOD, Fully Supervised |
| inbreast [77] | FSOD, Fully Supervised |
| liver-disease [88] | FSOD, Fully Supervised |
| nih-xray [53] | FSOD, Fully Supervised |
| spinefrxnormalvindr [128] | FSOD, Fully Supervised |
| stomata-cells [13] | FSOD, Fully Supervised |
| train [149] | FSOD, Fully Supervised |
| ufba-425 [122] | FSOD, Fully Supervised |
| urine-analysis1 [131] | FSOD, Fully Supervised |
| x-ray-id [111] | FSOD, Fully Supervised |
| xray [139] | FSOD, Fully Supervised |

| Aerial | Link |
|---|---|
| aerial-airport [61] | FSOD, Fully Supervised |
| aerial-cows [61] | FSOD, Fully Supervised |
| aerial-sheep [115] | FSOD, Fully Supervised |
| apoce-aerial-photographs-for-object-detection-of-construction-equipment [114] | FSOD, Fully Supervised |
| electric-pylon-detection-in-rsi [112] | FSOD, Fully Supervised |
| floating-waste [60] | FSOD, Fully Supervised |
| human-detection-in-floods [92] | FSOD, Fully Supervised |
| sssod [69] | FSOD, Fully Supervised |
| uavdet-small [110] | FSOD, Fully Supervised |
| wildfire-smoke [95] | FSOD, Fully Supervised |
| zebrasatasturias [123] | FSOD, Fully Supervised |

| Sports | Link |
|---|---|
| actions [31] | FSOD, Fully Supervised |
| aerial-pool [2] | FSOD, Fully Supervised |
| ball [72] | FSOD, Fully Supervised |
| bibdetection [134] | FSOD, Fully Supervised |
| football-player-detection [129] | FSOD, Fully Supervised |
| lacrosse-object-detection [118] | FSOD, Fully Supervised |

| Other | Link |
|---|---|
| buoy-onboarding [109] | FSOD, Fully Supervised |
| car-logo-detection [79] | FSOD, Fully Supervised |
| clashroyalechardetector [27] | FSOD, Fully Supervised |
| cod-mw-warzone [76] | FSOD, Fully Supervised |
| countingpills [41] | FSOD, Fully Supervised |
| everdaynew [56] | FSOD, Fully Supervised |
| flir-camera-objects [6] | FSOD, Fully Supervised |
| halo-infinite-angel-videogame [8] | FSOD, Fully Supervised |
| mahjong [99] | FSOD, Fully Supervised |
| new-defects-in-wood [48] | FSOD, Fully Supervised |
| orionproducts [102] | FSOD, Fully Supervised |
| pill [101] | FSOD, Fully Supervised |
| soda-bottles [12] | FSOD, Fully Supervised |
| taco-trash-annotations-in-context [135] | FSOD, Fully Supervised |
| the-dreidel-project [80] | FSOD, Fully Supervised |

