# OpenReview forum: "Roboflow100-VL: A Multi-Domain Object Detection Benchmark for Vision-Language Models"
_NeurIPS.cc/2025/Datasets_and_Benchmarks_Track — NeurIPS 2025 Datasets and Benchmarks Track poster_

### Official Review · Reviewer_ZDTB · 2025-06-19

**Rating:** 5
**Confidence:** 4

**Summary:**

The paper provides a key assessment tool for the practical application of VLMs (such as medical diagnosis and industrial quality inspection) by constructing a multi-domain detection benchmark. The experimental design is rigorous, and the conclusions have practical guidance significance. However, there are still some areas that need improvement.

**Dataset Code Accessibility:**

Yes

**Ethical Considerations:**

No, there are no or only very minor ethics concerns

**Final Justification:**

In the author's response, detailed comparisons between the RF100-VL dataset and other relevant datasets have been added, with a particular emphasis on the rich textual descriptions in the RF100-VL dataset. Additionally, visualization content has been included, the strategy for handling few-shot split has been explained, and the causes of failed cases have been analyzed. In summary, the author has provided detailed and comprehensive responses to all issues raised in the review comments, offering positive feedback or modifications. I consider the author's method design to be logically sound, with experiments conducted in a rigorous and thorough manner.

**Limitations Weaknesses:**

1. Insufficient depth of comparison with existing benchmark datasets: Although the paper mentions some existing visual-language understanding benchmark datasets, the detailed comparison between RoboFlow100-VL and these datasets in terms of data scale, diversity, and task difficulty is not sufficiently in-depth, making it difficult for readers to fully understand its unique advantages and positioning. The authors could enhance the relevant sections with more comprehensive comparative analyses against existing datasets to highlight the irreplaceable role of RoboFlow100-VL in evaluating model generalization capabilities.
2. Insufficient visualization in the experimental section: The paper primarily presents experimental results in tabular form, lacking visual analyses of key experimental outcomes, such as comparison diagrams of model detection performance across different datasets or trend diagrams of model performance changes during few-shot learning.
3. The few-shot setting uses a 10-shot split, but the sample selection strategy (e.g., random sampling vs. stratified sampling) is not specified, which may lead to class imbalance affecting the results. It is recommended to clearly specify the statistical information of the few-shot split.
4. The fine-tuning strategies for different models (e.g., Detic uses federated loss, while YOLO does not use frequency priors) lack a unified baseline. It is recommended to adopt a standardized fine-tuning protocol to ensure comparability between models.
5. The paper only compares performance under different data modalities without exploring the fundamental causes of model failures (e.g., semantic ambiguity, modal mismatch). For example, whether medical image detection failures are due to a lack of anatomical knowledge, or whether industrial data suffers from feature extraction difficulties due to imaging noise. It is recommended to add an error analysis section to reveal the causes of failures through visualization or case studies.

**Strengths Contributions:**

1. Curated across-domain datasets (564 categories, 164,000 images, 1.35 million annotations) covering tasks rarely seen in internet pre-training, such as medical imaging (e.g., liver fibrosis) and industrial defect detection. These datasets significantly exceed the scope of traditional VLMs (e.g., COCO, ImageNet), providing a new dimension for evaluating model generalization capabilities and addressing the insufficient coverage of rare concepts in existing benchmarks (e.g., RefCOCO, OdinW).
2. A comprehensive evaluation of various advanced VLMs was conducted under zero-shot, few-shot, semi-supervised, and fully supervised settings. This comprehensive evaluation not only demonstrates the performance of different models under various data conditions but also provides rich data support for VLM researchers.
3. Addressing the issue of poor generalization ability of current VLM models when faced with long-tail distribution data, the study proposes using multi-modal annotation to simulate the concept alignment process of human annotators, guiding the model to utilize contextual information rather than relying solely on category names. This design provides a more practical evaluation paradigm for few-shot learning.

---

> ### Author Rebuttal · Authors · 2025-07-31
>
> Reviewer ZDTB appreciates our “comprehensive evaluation”, which “not only demonstrates  the performance of different models under various data conditions but also provides rich data support for VLM researchers”. We thank them for their review and address their concerns below.
>
> ### **Dataset comparison table**
> We present a detailed comparison of RF100-VL with related datasets below. Notably, RF100-VL is the only dataset to support rich textual descriptions and evaluates models in the zero-shot, few-shot, semi-supervised, and fully-supervised data regimes. We will include this in our updated draft.
>
> | **Dataset**           | **# Datasets** | **# Images** | **# Annotations** | **# Classes** | **Rich Textual Descriptions** | **Multi-spectral Imagery** | **Zero-Shot Evaluation** | **Few-Shot Evaluation** | **Semi-Supervised Evaluation** | **Fully-Supervised Evaluation** | **Image Source**  |
> |-----------------------|----------------|--------------|-------------------|---------------|-------------------------------|----------------------------|--------------------------|-------------------------|--------------------------------|---------------------------------|-------------------|
> | Roboflow100-VL (Ours) | 100            | 164,149      | 1,355,491         | 564           | Yes                           | Yes                        | Yes                      | Yes                     | Yes                            | Yes                             | Roboflow Universe |
> | Roboflow100           | 100            | 224,714      | 1,319,307         | 805           | No                            | Yes                        | No                       | No                      | No                             | Yes                             | Roboflow Universe |
> | COCO                  | 1              | 328,000      | 2,500,000         |  91           | No                            | No                         | No                       | No                      | No                             | Yes                             | Flickr            |
> | LVIS (v0.5)           | 1              | 82,000       | 745,000           | 1230          | No                            | No                         | No                       | No                      | No                             | Yes                             | COCO              |
> | Objects365            | 1              | 638,000      | 10,101,000        | 365           | No                            | No                         | No                       | No                      | No                             | Yes                             | Flickr            |
> | OdinW                 | 35             | 152,384      | 1,073,455         | 314           | No                            | No                         | Yes                      | Yes                     | No                             | Yes                             | Roboflow Universe |
>
> ### **Adding more visuals**
> Thanks for the suggestion! We will add more visuals highlighting the performance of different models across few-shot performance across datasets.
>
> ### **Selecting few-shot splits**
> We follow the sample selection strategy proposed in Wang et. al. [1] (cf. Fig 4). To construct a $K$-shot dataset, we select a target class c and an image at random. If the total annotations for class $C$ in the image are less than or equal to $K$, we add the image to our dataset. We repeat this process for all classes until we have exactly $K$ annotations per class. This ensures that each category has exactly 10 examples. We will make this more clear in the updated draft.
>
> [1] Frustratingly Simple Few-Shot Object Detection. Wang et. al. ICML 2020
>
> ### **Clarification on fine-tuning strategies**
> Thanks for the suggestion! We will modify our Detic results to avoid using frequency priors in the updated draft. Although Madan et. al. [2] proposes using frequency priors based on global dataset statistics, we avoid introducing such priors since this information is not freely available in the few-shot setup. Moreover, we find that adding frequency priors does not significantly impact model performance in RF100-VL.
>
> [2] Revisiting Few-Shot Object Detection with Vision-Language Models. Madan et. al. NeurIPS D&B 2024
>
> ### **Analysis of failure cases**
> We can infer the causes of model failure by comparing the relative performance of standard object detectors (e.g. YOLOv8) with VLMs (e.g. Detic) for few-shot object detection. Importantly, unlike Detic, YOLOv8 is not pre-trained on large-scale datasets and is not promptable with class names. Therefore, when both models achieve low performance, we can attribute model failures to difficulties in feature extraction. In contrast, when YOLOv8 performs well, but Detic performance suffers, we can attribute model failures to semantic ambiguity.
>
> Using this heuristic, we analyze YOLOv8m w/ Federated Loss and Detic w/ Federated Loss because they have similar overall performance on RF100-VL (cf. Table 2). Notably, we find that Detic achieves 19.6 AP on Documents, while YOLOv8m achieves 25.9 AP. This suggests that these datasets contain many semantically ambiguous classes. Similarly, Detic achieves 8.5 AP on Medical while YOLOv8m achieves 15.9 AP. In contrast, we posit that datasets where Detic outperforms YOLOv8 (like Flora & Fauna and Sports) are semantically unambiguous and more similar to Detic’s pre-training. We will add this discussion in the updated draft.

---

> > ### Comment · Reviewer_ZDTB · 2025-08-08
> > **Thank you for your review**
> >
> > This review is concise and sufficient. Thank you for your contribution

---

### Official Review · Reviewer_QrMH · 2025-06-30

**Rating:** 5
**Confidence:** 4

**Summary:**

This paper presents a multi-domain object detection benchmark designed for vision-language models. It includes concepts that are not typically found in internet-scale datasets, curated from 100 challenging and diverse datasets with multimodal annotations. State-of-the-art models are evaluated across various settings, including zero-shot, few-shot, semi-supervised, and fully supervised scenarios. Overall, the proposed benchmark, Roboflow100-VL, is a large-scale dataset that has the potential to serve as a standard testbed for future VLMs and MLLMs.

**Dataset Code Accessibility:**

Yes

**Ethical Considerations:**

No, there are no or only very minor ethics concerns

**Final Justification:**

The authors address my concerns. It is now a more positive paper for the dataset track.

**Limitations Weaknesses:**

1. The paper mentions that the dataset contains images "not typically found in internet-scale datasets" or refers to it as "out-of-distribution," and claims it can evaluate the generalizability of VLMs to such tasks. However, these statements remain vague. The authors should provide more concrete descriptions or examples of what makes the dataset atypical and out of distribution. For instance, from a medical perspective, does this mean the data contains rare disease cases, or are these image types or object classes completely absent from VLM training? Clarifying this point would help the audience better understand the novelty and significance of the benchmark.

2.  It would be helpful if the authors included a comparison table summarizing how their dataset differs from or improves upon existing publicly available object detection datasets. This table could compare key attributes such as image resolution, number of classes, and dataset size. Such a comparison would offer a clearer and more immediate understanding of the dataset’s unique contributions and its positioning.

**Strengths Contributions:**

1. The dataset is large-scale, containing over 564 object classes and 164,149 images. All images are well-organized and thoroughly annotated, making it an accessible and practical benchmark for the academic community.
2. The annotation protocol and baseline model code are publicly available on GitHub, enhancing the reproducibility and usability of the benchmark across the research community. A challenge has already been conducted based on this benchmark, indicating its early adoption and utility in real-world applications.

---

> ### Author Rebuttal · Authors · 2025-07-31
>
> Reviewer QrMH comments that our CVPR challenge indicates RF100-VL’s “early adoption and utility in real world applications.” We thank them for their review and address their concerns below.
>
> ### **What makes datasets in RF100-VL "out of distribution"?**
> We use the term “out of distribution” to highlight domains in which state-of-the-art VLMs perform significantly worse than on contemporary benchmarks. As shown in Table 1, Qwen 2.5VL and Gemini 2.5 Pro achieve more than 30 AP on OdinW-13, but score less than 10 AP on RF100-VL. We carefully curate RF100-VL such that it cannot be solved by simply prompting state-of-the-art models with class names. Specifically, we include datasets where classes are labeled using scientific names (e.g. liver fibrosis and steatosis), acronyms (e.g. DIP and MCP), context-dependent names (e.g. detecting a block vs. set in the context of volleyball), material properties (e.g. paper vs. soft plastic), and diverse imaging modalities (cf. Fig. 2). Notably, contemporary datasets like COCO and OdinW do not include such diverse imaging modalities or such challenging class names. We will clarify in our updated draft.
>
> ### **Dataset comparison table**
> We present a detailed comparison of RF100-VL with related datasets below. Notably, RF100-VL is the only dataset to support rich textual descriptions and evaluates models in the zero-shot, few-shot, semi-supervised, and fully-supervised data regimes. We will include this in our updated draft.
>
> | **Dataset**           | **# Datasets** | **# Images** | **# Annotations** | **# Classes** | **Rich Textual Descriptions** | **Multi-spectral Imagery** | **Zero-Shot Evaluation** | **Few-Shot Evaluation** | **Semi-Supervised Evaluation** | **Fully-Supervised Evaluation** | **Image Source**  |
> |-----------------------|----------------|--------------|-------------------|---------------|-------------------------------|----------------------------|--------------------------|-------------------------|--------------------------------|---------------------------------|-------------------|
> | Roboflow100-VL (Ours) | 100            | 164,149      | 1,355,491         | 564           | Yes                           | Yes                        | Yes                      | Yes                     | Yes                            | Yes                             | Roboflow Universe |
> | Roboflow100           | 100            | 224,714      | 1,319,307         | 805           | No                            | Yes                        | No                       | No                      | No                             | Yes                             | Roboflow Universe |
> | COCO                  | 1              | 328,000      | 2,500,000         |  91           | No                            | No                         | No                       | No                      | No                             | Yes                             | Flickr            |
> | LVIS (v0.5)           | 1              | 82,000       | 745,000           | 1230          | No                            | No                         | No                       | No                      | No                             | Yes                             | COCO              |
> | Objects365            | 1              | 638,000      | 10,101,000        | 365           | No                            | No                         | No                       | No                      | No                             | Yes                             | Flickr            |
> | OdinW                 | 35             | 152,384      | 1,073,455         | 314           | No                            | No                         | Yes                      | Yes                     | No                             | Yes                             | Roboflow Universe |

---

> > ### Comment · Reviewer_QrMH · 2025-08-05
> >
> > Thank you for the explanation and new results. I am reaffirming my positive rating.

---

### Official Review · Reviewer_u8Zq · 2025-07-01

**Rating:** 5
**Confidence:** 4

**Summary:**

This paper presents Roboflow100-VL, a large-scale benchmark comprising 100 multi-domain object detection datasets, aimed at evaluating the generalization ability of vision-language models (VLMs) on out-of-distribution tasks. The benchmark is notable for including challenging concepts rarely seen in standard pretraining data, such as medical terms, industrial materials, and non-RGB imaging modalities.\
A key contribution is the introduction of multi-modal annotator instructions, pairing visual exemplars with textual descriptions to support few-shot learning and better concept alignment. The authors evaluate models under zero-shot, few-shot, semi-supervised, and fully-supervised settings, finding that current state-of-the-art models perform poorly in low-data regimes—particularly generalist MLLMs—while specialist detectors such as GroundingDINO perform more robustly.\
Overall, the benchmark highlights critical gaps in current VLM capabilities and provides a valuable resource for advancing research in vision-language alignment under limited supervision.

**Dataset Code Accessibility:**

Yes

**Ethical Considerations:**

No, there are no or only very minor ethics concerns

**Final Justification:**

Appreciate the detailed rebuttal. The author has solved all my problems, and I have decided to raise my rating.

**Limitations Weaknesses:**

1.Although the paper introduces multi-modal annotator instructions, many are generated by GPT-4o and only lightly edited. As acknowledged in Sec. 4.4, these instructions may lack domain-specific detail—especially for complex fields like medicine. Prior work, such as Meta-ADD: A meta-learning based pre-trained model for concept drift active detection, emphasizes the importance of active human-guided alignment under concept drift, suggesting that more robust human-in-the-loop instruction generation could improve both instruction quality and few-shot performance.\
2.The datasets are sourced from Roboflow Universe, and while manual validation is mentioned, it is unclear how exhaustive or consistent the quality control was across all 100 datasets. Further transparency in annotation correction protocols or releasing quality assessment metrics would enhance trust in the benchmark.\
3.As shown in Table 2, multi-modal prompts (text + image) sometimes perform worse than simpler class name prompts. For instance, Gemini 2.5 Pro shows degraded performance with full instructions. This inconsistency may stem from prompt format mismatch or instruction overload. A more adaptive prompt design or fine-tuning strategy could be explored.

**Strengths Contributions:**

1.The paper introduces Roboflow100-VL, a benchmark of 100 diverse object detection datasets focused on out-of-distribution concepts, including scientific terms, acronyms, material properties, and non-RGB modalities. This significantly expands the evaluation scope beyond prior datasets like COCO and ODinW.\
2.A key contribution is the use of annotator-style instructions that combine few-shot visual examples with textual descriptions per class. This setup mimics human annotation processes and enables realistic few-shot evaluation.\
3.The authors clearly position RF100-VL relative to prior benchmarks, showing that existing datasets fail to capture the challenges presented here (Sec. 2, Sec. 4.2). The curated dataset avoids class-name shortcuts and emphasizes hard examples requiring contextual understanding.\
4.The paper is well-structured and easy to follow. Figures and tables are informative and support the claims effectively. Code and data are publicly available, and the benchmark is accompanied by a CVPR 2025 challenge to drive adoption.

---

> ### Author Rebuttal · Authors · 2025-07-31
>
> Reveiwer u8Zq appreciates that RF100-VL “significantly expands the evaluation scope beyond prior datasets like COCO and ODinW”. We thank them for their review and address their concerns below.
>
> ### **Importance of human-in-the-loop refinement for annotator instructions**
> Thanks for highlighting this relevant paper! Motivated by the observation from Meta-ADD, we evaluate few-shot detection performance on RF20-VL using annotator instructions generated by GPT4o, Qwen 2.5-VL, Gemini 2.5 Pro, GPT4o with a human-in-the loop (our original instructions) using the prompt from Appendix J, and human written instructions. We evaluate the impact of instruction source on Qwen 2.5 VL and Gemini 2.5 Pro. Notably, we do not find a clear correlation between instruction source and downstream model performance. We find that Qwen 2.5 VL achieves better performance with annotator instructions from all sources compared to class names only, while Gemini 2.5 Pro performs worse with annotator instructions from all sources compared to class names only. Somewhat surprisingly, we find that instructions from GPT 4o with a human-in-the-loop performs the best on both Qwen 2.5VL and Gemini 2.5 Pro, beating human written instructions. Although prompting with annotator instructions yields inconsistent benefits, future work should explore novel ways of incorporating such rich contextual information. We will add this discussion to the updated draft.
>
> | **Instruction Source**                      | **Qwen 2.5VL** | **Gemini 2.5 Pro** |
> |---------------------------------------------|----------------|--------------------|
> | Class Names Only                            | 5.0            | 15.5               |
> | GPT4o Instructions                          | 6.3            | 5.3                |
> | Qwen 2.5 VL Instructions                    | 6.3            | 4.6                |
> | Gemini 2.5 Pro Instructions                 | 7.2            | 5.2                |
> | GPT-4o Instructions with Edits (Main Paper) | 7.2            | 6.8                |
> | Human Written Instructions                  | 6.5            | 4.2                |
>
> ### **Details on annotation quality**
> We hired external contractors to refine RF100-VL’s bounding box annotations according to a set of guidelines, described below. In total, 30 annotators spent 2168 hours validating annotations, with the authors performing additional quality control. Our annotation guidelines provide instructions for improving the quality of existing annotations across datasets. Our primary goal was to ensure consistent annotation style, with every possible instance of each class labeled by a single, tightly fitting bounding box.
>
> The annotation refinement process emphasizes the following corrections:
> - _Merged Bounding Boxes_: Annotators must ensure each object has its own bounding box, redrawing boxes that encompass multiple instances of an object.
> - _Incomplete Bounding Boxes_: Bounding boxes must fully contain the entire object. If an object extends beyond the box's boundaries, the box needs to be expanded to include the whole object.
> - _Missing Annotations_: It is crucial to identify and label every instance of each class within a dataset. This is the most challenging and important aspect of refinement. Annotators are advised to check the background for missing annotations.
> - _Incorrectly Labeled Objects_: Bounding boxes around objects that do not belong to the specified class must be removed.
> - _Wrong Class Names_: Annotators need to correct instances where class names are misassigned to objects (e.g., doors labeled as windows, or generic numerical labels instead of descriptive class names like "enemy" or "head").
> - _Duplicated Bounding Boxes_: If the same object has multiple bounding boxes, one of the duplicates must be removed.
>
> These instructions focus on correcting annotations within each dataset to achieve consistency. When inconsistencies are found in a dataset's labeling scheme, annotators are instructed to make a determination for consistency and ensure all images follow that scheme. We will include our full annotation guide in the updated draft.
>
> ### **Why do multi-modal prompts perform worse than class names?**
> We investigate why more descriptive prompts lead to worse detection performance by evaluating few-shot detection performance on RF20-VL using annotator instructions generated by GPT4o with a human-in-the loop (our original instructions) using the prompt from Appendix J, a shortened version of these instructions, and human-written instructions. On average, the class names only prompts had 31.75 words, the GPT4o instructions with a human-in-the-loop had 502 words, the shortened GPT4o instructions with a human-in-the-loop had 170.55 words, and the human instructions had 482.95 words. Somewhat surprisingly, we find that the length of the prompt does not correlate well with model performance. This suggests that the models are not fine-tuned to leverage such instructions, regardless of context length. We agree that future work should consider more adaptive prompt designs or fine-tuning strategies.
>
> | **Instruction Source**                      | **Qwen 2.5VL** | **Gemini 2.5 Pro** |
> |---------------------------------------------|----------------|--------------------|
> | Class Names Only                            | 5.0            | 15.5               |
> | GPT-4o Instructions with Edits (Main Paper) | 7.2            | 6.8                |
> | Shortened GPT-4o Instructions with Edits    | 5.8            | 5.4                |
> | Human Instructions                          | 6.5            | 4.2                |

---

> > ### Comment · Reviewer_u8Zq · 2025-08-06
> >
> > Thank you for the additional clarifications and experiments. Overall, I find your response convincing.
> >
> > GPT-4o with human edits performs best on both Qwen 2.5-VL and Gemini 2.5 Pro, however, on Gemini 2.5 Pro, instruction-augmented multimodal prompts still underperform simple class-name prompts, which does suggest model-specific sensitivity to prompt format/modality. On annotation consistency, the added human verification process and concrete guidelines largely address my earlier concerns. Regarding prompt “length”, I agree with your assessment: the problem appears to lie not in length, but in insufficient alignment/conditioning to leverage richer context.
> >
> > **I have two further requests. If the workload is heavy, please prioritize either one, I will increasing my score accordingly.**
> >
> > * In a few-shot setting (e.g., 5–10 shot), add a very lightweight “detection-oriented” post-processing/calibration pipeline for general-purpose MLLMs (confidence calibration plus unified NMS and score-threshold search) and compare against the raw outputs. This will help disentangle model capacity from missing interfaces/post-processing, and may clarify why multimodal instructions yield unstable gains.
> >
> > * Since length is not the key factor, the more important question is the information structure. Please decompose instructions into fixed components (e.g., precise definition; inclusion/exclusion rules, etc.) and run a component-wise ablation. Report marginal contributions separately on specialist detectors and general-purpose MLLMs; optionally inject mild noise or slightly misleading descriptions to test robustness and identify which components are stable versus fragile.

---

> > ### Author Response · Authors · 2025-08-07
> > **Follow Up on Comments**
> >
> > Dear Reviewer u8Zq,
> >
> > Thanks for following up! We present additional experimental results to address your insightful questions below.
> >
> > ### ***Impact of Detector Post-Processing***
> > Unlike specialist detectors, MLLMs directly predict bounding boxes without confidence scores, and do not leverage common detector post-processing techniques like NMS. We investigate the impact of such post-processing steps on MLLM detection accuracy with RF20-VL. First, we estimate the confidence score of each predicted bounding box with SigLIPv2. We compute the cosine similarity of the predicted class name text embedding and bounding box image crop embedding. Although one can prompt an MLLM to predict its own confidence scores in theory, recent work [1] demonstrates that MLLMs struggle to verbalize confidence estimates in practice. We expect that the challenging out-of-distribution classes in RF20-VL make directly verbalizing confidence estimates even more difficult. Further, modifying the MLLM prompt to predict confidences for each bounding box introduces additional sources of variance. Next, we run NMS per-class. Importantly, these post-processing steps can be applied to both zero-shot and few-shot prompted MLLMs. We find that adding confidence scores from SigLIP significantly improves performance for both Qwen 2.5VL and Gemini 2.5 Pro. Further, NMS seems to have a negligible impact, suggesting the LLMs implicitly learn to avoid making duplicate predictions.
> >
> > | **Model**             | **Qwen 2.5VL** | **Gemini 2.5 Pro** |
> > |-----------------------|----------------|--------------------|
> > | Instructions          | 7.2            | 6.8                |
> > |    + SigLIP Score     | 9.7            | 8.9                |
> > |    + NMS              | 9.8            | 8.9                |
> > | Instructions + Images | 7.1            | 10.7               |
> > |    + SigLIP Score     | 8.8            | 12.5               |
> > |    + NMS              | 8.9            | 12.5               |
> >
> > [1] Seeing is Believing, but How Much? A Comprehensive Analysis of Verbalized Calibration in Vision-Language Models. Xuan et. al. ArXiv 2025.
> >
> > ### ***Ablation on Impact of Instruction Parts***
> > We agree that this is an interesting ablation to evaluate the marginal contribution of different parts of the instruction. However, due to time constraints and API rate limiting, we are unable to provide empirical evidence at this time. We will explore this ablation in the updated version.

---

> > > ### Comment · Reviewer_u8Zq · 2025-08-08
> > >
> > > Thank you for the response. It has addressed my concerns, and I will raise my score.

---

### Official Review · Reviewer_fE7H · 2025-07-03

**Rating:** 4
**Confidence:** 4

**Summary:**

This paper introduces Roboflow100-VL (RF100-VL), a large-scale benchmark comprising 100 diverse object detection datasets designed to evaluate vision-language models (VLMs) on challenging, out-of-distribution concepts. The benchmark emphasizes multi-modal concept alignment through the use of few-shot visual examples and rich textual descriptions. The authors evaluate a range of state-of-the-art models under zero-shot, few-shot, semi-supervised, and fully-supervised settings. Experimental results show that current models perform poorly on RF100-VL compared to existing benchmarks, underscoring limitations in generalization capabilities. The paper further contributes curated annotation instructions and hosts a public challenge to encourage community engagement.

**Dataset Code Accessibility:**

Yes

**Ethical Considerations:**

No, there are no or only very minor ethics concerns

**Final Justification:**

I have carefully reviewed the authors’ rebuttal and considered the discussion among the reviewers. The authors have addressed my concerns, and therefore I will maintain my original score.

**Limitations Weaknesses:**

The empirical benefit of the proposed multi-modal annotator instructions remains inconclusive. As discussed in Lines 206–213, prompting with these instructions leads to inconsistent improvements—Qwen2.5-VL shows minor gains, while Gemini 2.5 Pro degrades. Since the annotator instructions are automatically generated by GPT-4o, their content may inherit model-specific biases or omit critical details. The paper would benefit from a more systematic evaluation of the impact of instruction quality and origin (e.g., human-written vs. model-generated), especially given that the benchmark’s central premise relies heavily on these instructions.

**Strengths Contributions:**

1. The paper clearly articulates a critical limitation of existing open-vocabulary detection benchmarks like ODinW, which often contain commonly seen categories and thus fail to properly assess the true zero-shot capabilities of VLMs.
2. By curating datasets with non-trivial class names (e.g., scientific terms and domain-specific acronyms), the proposed benchmark provides a more rigorous and realistic test of a model's ability to generalize to novel concepts.
3. The paper provides an in-depth comparative analysis between foundation vision-language models and specialized detectors across different data regimes, offering valuable insights for the community and motivating future research directions.

---

> ### Author Rebuttal · Authors · 2025-07-31
>
> Reviewer fE7H appreciates our “in-depth comparative analysis” that offers “valuable insights for the community”. We thank them for their review and address their concerns below.
>
> ### **Systemic evaluation of instruction quality and origin**
> Great suggestion! We evaluate few-shot detection performance on RF20-VL using annotator instructions generated by GPT4o, Qwen 2.5-VL, Gemini 2.5 Pro, GPT4o with a human-in-the loop (our original instructions) using the prompt from Appendix J, and human written instructions. We evaluate the impact of instruction source on Qwen 2.5 VL and Gemini 2.5 Pro. Notably, we do not find a clear correlation between instruction source and downstream model performance. We find that Qwen 2.5 VL achieves better performance with annotator instructions from all sources compared to class names only, while Gemini 2.5 Pro performs worse with annotator instructions from all sources compared to class names only. Somewhat surprisingly, we find that instructions from GPT 4o with a human-in-the-loop performs the best on both Qwen 2.5VL and Gemini 2.5 Pro, beating human written instructions. Although prompting with annotator instructions yields inconsistent benefits, future work should explore novel ways of incorporating such rich contextual information. We will add this discussion to the updated draft.
>
> | **Instruction Source**              | **Qwen 2.5VL** | **Gemini 2.5 Pro** |
> |---------------------------------------------|----------------|--------------------|
> | Class Names Only                            | 5.0            | 15.5               |
> | GPT4o Instructions                          | 6.3            | 5.3                |
> | Qwen 2.5 VL Instructions                    | 6.3            | 4.6                |
> | Gemini 2.5 Pro Instructions                 | 7.2            | 5.2                |
> | GPT-4o Instructions with Edits (Main Paper) | 7.2            | 6.8                |
> | Human Written Instructions                  | 6.5            | 4.2                |

---

> > ### Comment · Reviewer_fE7H · 2025-08-09
> >
> > Thank you to authors for the rebuttal and additional experiments. My concerns have been addressed. Please include the content in the rebuttal to your revised version of the paper.

---

### Author Response · Authors · 2025-08-07
**End of Discussion Period**

Dear Reviewers,

We'd like to thank you again for your feedback and discussion on our paper. Before the discussion period wraps up, we'd be happy to answer any additional questions you may still have and kindly request that you update your post-rebuttal scores. If we've addressed your concerns, please let us know! Please note that we cannot see any comments you may have shared in your rebuttal acknowledgement.

---

### Note · Authors · 2025-08-13

***Summary***

We are thrilled reviewers agree that Roboflow100-VL “significantly expands the evaluation scope beyond prior datasets like COCO and ODinW” (u8Zq), appreciate our “in-depth comparative analysis” (fE7H, ZDTB) that offers “valuable insights for the community” (fE7H, ZDTB) and recognize that our successful CVPR challenge suggests strong “early adoption and utility in real world applications” (QrMH). We thank all reviewers for their feedback. We summarize key reviewer concerns and our responses below.

***Major Reviewer Concerns***

- fE7H and u8Zq request systematic evaluation of annotation instruction origin and quality on MLLM performance. We provide this analysis in the rebuttal. Notably, we do not find a clear correlation between instruction source and downstream model performance.
- QrMH and ZDTB request a dataset comparison table to highlight the key differences between RF100-VL and prior datasets. We provide this table in the rebuttal. RF100-VL uniquely provides rich annotator instructions and evaluates models in zero-shot, few-shot, semi-supervised, and fully-supervised data regimes.
- ZDTB requests an analysis of detector failure cases to identify trends and potential causes. We provide this analysis in the rebuttal. We highlight that open-vocabulary detectors like Detic performs worse than closed-vocabulary detectors like YOLOv8 on the Documents and Medical super-categories, suggesting that these datasets contain many semantically ambiguous classes. In contrast, Detic outperforms YOLOv8 on Flora & Fauna and Sports, suggesting that these datasets classes are semantically unambiguous and more similar to Detic’s pre-training.

All reviewers acknowledge that we have sufficiently addressed their concerns. We will be sure to include these additional experiments in our updated version.

---

### Decision · Program_Chairs · 2025-09-18

**Decision:**

Accept (poster)

**Comment:**

This paper introduces Roboflow100-VL, a large-scale, multi-domain object detection benchmark designed to evaluate the generalization capabilities of Vision-Language Models (VLMs) on new concepts with a few visual examples and rich textual descriptions. The authors conducted extensive evaluations across multiple settings using a variety of state-of-the-art models. A key finding is the significant performance gap observed by using many modern VLMs in zero-shot scenarios. This result highlights the pressing need for few-shot concept alignment research.

The paper received strong reviews, with three "Accept" and one "Borderline Accept".

Strengths: The benchmark significantly expands the evaluation scope beyond prior datasets, provides an in-depth comparative analysis across different data regimes, offers valuable insights for the community, and motivates future research directions.

Weaknesses: The empirical benefit of the multimodal instructions was initially unclear. The initial version lacked a detailed comparison with existing datasets and a thorough analysis of model failure cases.

In the rebuttal and author-reviewer discussion phases, the authors provided a new empirical evaluation of the instructions, added a comprehensive comparison, and performed a detailed failure case analysis. Therefore, above these concerns are well addressed.

The paper is strongly recommended for acceptance. The authors' rebuttal was highly effective, directly addressing major concerns. This work holds significant importance in advancing VLMs research on new concept alignment in low-data regimes. The authors' thoroughness in responding to feedback, combined with the significant contribution of the dataset to the VLM community, makes this an excellent submission.

===== FINAL UPDATE FROM DB Track PCs ====

The final decision for this paper has been taken by the program chairs after consultation with the SACs. All Senior Area Chairs have ranked papers according to the feedback from the AC during the review process. We decided to leave the original meta-review to reflect the opinion of the AC in light of the initial discussions with reviewers and SAC.